# Sample Efficient Multiple-policy Evaluation in Reinforcement Learning

## Abstract

We study the multiple-policy evaluation problem where we are given a set of $K$ policies and the goal is to evaluate their performance (expected total reward over a fixed horizon) to an accuracy $\epsilon$ with probability at least $1 - \delta$. We propose a sample-efficient algorithm named CAESAR for this problem. Our approach is based on computing an approximate optimal offline sampling distribution and using the data sampled from it to perform the simultaneous estimation of the policy values. CAESAR has two phases. In the first we produce coarse estimates of the visitation distributions of the target policies at a low order sample complexity rate that scales with $\tilde{O}(\frac{1}{\epsilon})$. In the second phase, we approximate the optimal offline sampling distribution and compute the importance weighting ratios for all target policies by minimizing a step-wise quadratic loss function inspired by the DualDICE (Nachum et al., 2019) objective. Up to low order and logarithmic terms CAESAR achieves a sample complexity $\tilde{O}\left( \frac{H^4}{\epsilon^2} \sum_{h=1}^{H} \min_{\mu_h} \max_{k \in [K]} \sum_{s,a} \frac{(d_h^{\pi^k}(s,a))^2}{\mu_h(s,a)} \right)$, where $d^\pi$ is the visitation distribution of policy $\pi$, $\mu$ is the sampling distribution, and $H$ is the horizon.

## 1 Introduction

It is rumored that sometime before the fateful day of March 15, 44 BC, a soothsayer warned Caesar of the impending dangers awaiting him at the Senate house. "Beware the Ides of March," the soothsayer is said to have uttered—a warning that Caesar famously ignored. Immortalized in Shakespeare's play "Julius Caesar," this cautionary tale reminds us of the dangers of misjudgment and the unforeseen consequences of strategic decisions. In *Reinforcement Learning (RL)*, the stakes, while not mortal, are still significant as agents navigate complex environments to optimize their actions.

This paper delves into the problem of policy evaluation, a fundamental problem in RL (Sutton & Barto, 2018) of which the goal is to estimate the expected total rewards of a given policy. This process serves as an integral component in various RL methodologies, such as policy iteration and policy gradient approaches (Sutton et al., 1999), wherein the current policy undergoes evaluation followed by potential updates. Policy evaluation is also paramount in scenarios where prior to deploying a trained policy, thorough evaluation is imperative to ensure its safety and efficacy.

Broadly speaking there exist two scenarios where the problem of policy evaluation has been considered, known as *online* and *offline* data regimes. In online scenarios, a learner is interacting sequentially with the environment and is tasked with using its online deployments to collect helpful data for policy evaluation. The simplest method for online policy evaluation is Monte-Carlo estimation (Fonteneau et al., 2013). One can collect multiple trajectories by following the target policy, and use the empirical mean of the rewards as the estimator. These on-policy methods typically require executing the policy we want to estimate which may be unpractical or dangerous in many cases. For example, in a medical treatment scenario, implementing an untrustworthy policy can cause unfortunate consequences (Thapa et al., 2005). In these cases, offline policy evaluation may be preferable. In the offline case, the learner has access to a batch of data and is tasked with using it in the best way possible to estimate the value of a target policy. Many works focus on this problem leveraging various techniques, such as importance-sampling, model-based estimation, and doubly-robust estimators (Yin & Wang, 2020; Jiang & Li, 2016; Yin et al., 2021; Xie et al., 2019; Li et al., 2015).

Motivated by applications where one has multiple policies to evaluate, e.g., multiple policies trained using different hyperparameters, Dann et al. (2023) considered multiple-policy evaluation which aims to estimate the performance of a set of $K$ target policies instead of a single one. At first glance, multiple-policy evaluation does not pose challenges beyond single-policy evaluation since one can always use $K$ instances of single-policy evaluation to evaluate the $K$ policies. However, since the sample complexity of this procedure scales linearly as a function of $K$ this can be extremely sample-inefficient as it neglects potential similarities among the $K$ target policies.

Dann et al. (2023) proposed an on-policy algorithm that leverages the similarity among target policies based on an idea related to trajectory synthesis (Wang et al., 2020). The basic technique is that if more than one policy take the same action at a certain state, then only one sample is needed at that state which can be reused to synthesize trajectories for these policies. Their algorithm achieves an instance-dependent sample complexity which gives much better results when target policies have many overlaps.

In the context of single off-policy evaluation, the theoretical guarantees depend on the overlap between the offline data distribution and the visitations of the evaluated policy (Xie et al., 2019; Yin & Wang, 2020; Duan et al., 2020). These coverage conditions, which ensure that the data logging distribution (Xie et al., 2022) adequately covers the state space, are typically captured by the ratio between the densities corresponding to the offline data distribution and the policy to evaluate, also known as visitation ratios.

A single offline dataset can be used to evaluate multiple policies simultaneously. The policy evaluation guarantees will be different for each of the policies in the set, depending on how much overlap the offline distribution has with the policy visitation distributions. These observations inform an approach to the multiple policy evaluation problem different from (Dann et al., 2023) that can also leverage the policy visitation overlap in a meaningful way. Our algorithm is based on the idea of designing a behavior distribution with enough coverage of the target policy set. Once this distribution is computed, i.i.d. samples from the behavior distribution can be used to estimate the value of the target policies using ideas inspired in the offline policy optimization literature. Our algorithms consist of two phases:

1. Build coarse estimators of the policy visitation distributions and use them to compute a mixture policy that achieves a low visitation ratio with respect to all $K$ policies to evaluate.

2. Sample from this approximately optimal mixture policy and use these to construct mean reward estimators for all $K$ policies.

Coarse estimation of the visitation distributions up to constant multiplicative accuracy can be achieved at a cost that scales linearly, instead of quadratically with the inverse of the accuracy parameter (see Section 4.1) and polynomially in parameters such as the size of the state and action spaces, and the logarithm of the cardinality of the policy evaluation set. We propose the MARCH or *Multi-policy Approximation via Ratio-based Coarse Handling* algorithm (see Algorithm 3) for coarse estimation of the visitation distributions. Estimating the policy visitation distributions up to multiplicative accuracy is enough to find an approximately optimal behavior distribution that minimizes the maximum visitation ratio among all policies to estimate (see Section 4.2). The samples generated from this behavior distribution are used to estimate the target policy values via importance weighting. Since the importance weights are not known to sufficient accuracy, we propose the IDES or *Importance Density Estimation* algorithm (see Algorithm 1) for estimating these distribution ratios by minimizing a series of loss functions inspired by the DualDICE (Nachum et al., 2019) method (see Section 4.3). Combining these steps we arrive at our main algorithm (CAESAR) or *Coarse and Adaptive EStimation with Approximate Reweighing* algorithm (see Algorithm 2) that achieves a high probability finite sample complexity for the problem of multi-policy evaluation.

## 2 RELATED WORK

There is a rich family of off-policy estimators for policy evaluation (Liu et al., 2018; Jiang & Li, 2016; Dai et al., 2020; Feng et al., 2021; Jiang & Huang, 2020). But none of them is effective in our setting. Importance-sampling is a simple and popular method for off-policy evaluation but suffers exponential variance in the horizon (Liu et al., 2018). Marginalized importance-sampling has been proposed to get rid of the exponential variance. However, existing works all focus on

function approximations which only produce approximately correct estimators (Dai et al., 2020) or are designed for the infinite-horizon case (Feng et al., 2021). The doubly robust estimator (Jiang & Li, 2016; Hanna et al., 2017; Farajtabar et al., 2018) also solves the exponential variance problem, but no finite sample result is available. Our algorithm is based on marginalized importance-sampling and addresses the above limitations in the sense that it provides non-asymptotic sample complexity results and works for finite-horizon *Markov Decision Processes (MDPs)*.

Another popular estimator is called model-based estimator, which evaluates the policy by estimating the transition function of the environment (Dann et al., 2019; Zanette & Brunskill, 2019). Yin & Wang (2020) provide a similar sample complexity to our results. However, there are some significant differences between their result and ours. First, our sampling distribution; calculated based on the coarse distribution estimator, is optimal. Second, our sample complexity is non-asymptotic while their result is asymptotic. Third, the true distributions appearing in our sample complexity can be replaced by known distribution estimators without inducing additional costs, that is, we can provide a known sample complexity while their result is always unknown since we do not know the true visitation distributions of target policies.

The work that most aligns with ours is Dann et al. (2023), which proposed an on-policy algorithm based on the idea of trajectory synthesis. The authors propose the first instance-dependent sample complexity analysis of the multiple-policy evaluation problem. Our algorithm offers a different perspective which uses off-policy evaluation based on importance-weighting and achieves a competitive sample complexity with simpler techniques and analysis.

In concurrent work, Amortila et al. (2024b) proposed an exploration objective for downstream reward maximization, similar to our goal of computing an optimal sampling distribution. However, our approach utilizes coarse distribution estimators to approximate the objective, which is a novel technique and an important contribution of our work.

Our algorithm also uses some techniques modified from other works which we summarize here. DualDICE is a technique for estimating distribution ratios by minimizing some loss functions proposed by (Nachum et al., 2019). Our algorithm IDES is built on this idea. However, these are significant differences, which are discussed in detail in Section 4.3. Besides, we utilize stochastic gradient descent algorithms and their convergence rate for strongly-convex and smooth functions from the optimization literature (Hazan & Kale, 2011). Finally, we adopt the Median of Means estimator (Minsker, 2023) to convert in-expectation results to high-probability results.

## 3 PRELIMINARIES

**Notations** We denote the set $\{1, 2, \ldots, N\}$ by $[N]$. $\{X_n\}_{n=1}^N$ represents the set $\{X_1, X_2, \ldots, X_N\}$. $\mathbb{E}_\pi$ denotes the expectation over the trajectories induced by policy $\pi$. $\tilde{O}$ hides constants, logarithmic and lower-order terms. We use $\mathbb{V}[X]$ to represent the variance of random variable $X$. $\Pi_{det}$ is the set of all deterministic policies and $\text{conv}(\mathcal{X})$ represents the convex hull of the set $\mathcal{X}$.

**Reinforcement learning framework** We consider episodic tabular Markov Decision Processes (MDPs) defined by a tuple $\{\mathcal{S}, \mathcal{A}, H, \{P_h\}_{h=1}^H, \{r_h\}_{h=1}^H, \nu\}$, where $\mathcal{S}$ and $\mathcal{A}$ represent the state and action spaces, respectively, with $S$ and $A$ denoting the respective cardinality of these sets. $H$ is the horizon which defines the number of steps the agent can take before the end of an episode. $P_h(\cdot|s, a) \in \Delta\mathcal{S}$ is the transition function which represents the probability of transitioning to the next state if the agent takes action $a$ at state $s$. And $r_h(s, a)$ is the reward function, accounting for the reward the agent can collect by taking action $a$ at state $s$. In this work, we assume that the reward is deterministic and bounded $r_h(s, a) \in [0, 1]$, which is consistent with prior work Dann et al. (2023). We denote the initial state distribution by $\nu \in \Delta\mathcal{S}$.

A policy $\pi = \{\pi_h\}_{h=1}^H$ is a mapping from the state space to the probability distribution space over the action space. $\pi_h(a|s)$ denotes the probability of taking action $a$ at state $s$ and step $h$. The value function $V_h^\pi(s)$ of a policy $\pi$ is the expected total reward the agent can receive by starting from step $h$, state $s$, and following the policy $\pi$, i.e., $V_h^\pi(s) = \mathbb{E}_\pi[\sum_{l=h}^H r_l|s]$. The performance $J(\pi)$ of a policy $\pi$ is defined as the expected total reward the agent can obtain. By the definition of the value function, we have $J(\pi) = V_1^\pi(s|s \sim \nu)$. For simplicity, in the following context, we use $V_1^\pi$ to denote $V_1^\pi(s|s \sim \nu)$.

The state visitation distribution $d_h^\pi(s)$ of a policy $\pi$ represents the probability of reaching state $s$ at step $h$ if the agent starts from a state sampled from the initial state distribution $\nu$ at step $l = 1$ and follows policy $\pi$ subsequently, i.e., $d_h^\pi(s) = \mathbb{P}[s_h = s | s_1 \sim \nu, \pi]$. Similarly, the state-action visitation distribution $d_h^\pi(s, a)$ is defined as $d_h^\pi(s, a) = d_h^\pi(s)\pi(a|s)$. Based on the definition of the visitation distribution, the performance of policy $\pi$ can also be expressed as $J(\pi) = V_1^\pi = \sum_{h=1}^{H} \sum_{s,a} d_h^\pi(s, a) r_h(s, a)$.

**Multiple-policy evaluation problem setup**  In multiple-policy evaluation, we are given a set of known policies $\{\pi^k\}_{k=1}^{K}$ and a pair of factors $\{\epsilon, \delta\}$. The objective is to evaluate the performance of these given policies such that with probability at least $1 - \delta$, $\forall \pi \in \{\pi^k\}_{k=1}^{K}$, $|\hat{V}_1^\pi - V_1^\pi| \le \epsilon$, where $\hat{V}_1^\pi$ is the performance estimator.

Dann et al. (2023) proposed an algorithm based on the idea of trajectory stitching and achieved an instance-dependent sample complexity,

$$\tilde{O}\left(\frac{H^2}{\epsilon^2}\mathbb{E}\left[\sum_{(s,a)\in\mathcal{K}^{1:H}}\frac{1}{d^{max}(s)}\right] + \frac{SH^2K}{\epsilon}\right), \tag{1}$$

where $d^{max}(s) = \max_{k\in[K]} d^{\pi^k}(s)$ and $\mathcal{K}^h \subseteq \mathcal{S} \times \mathcal{A}$ keeps track of which state-action pairs at step $h$ are visited by target policies in their trajectories.

Another way to reuse samples for evaluating different policies is to estimate the model. Based on the model-based estimator proposed by Yin & Wang (2020), an asymptotic convergence rate can be derived,

$$\sqrt{\frac{H}{n}} \cdot \sqrt{\sum_{h=1}^{H}\mathbb{E}_{\pi^k}\left[\frac{d^{\pi^k}(s_h, a_h)}{\mu(s_h, a_h)}\right]} + o\left(\frac{1}{\sqrt{n}}\right), \tag{2}$$

where $\mu$ is the distribution of the offline dataset and $n$ is the number of trajectories in this dataset. Though, it looks similar to our results, we have claimed in the Section 2 that there are significant differences.

## 3.1 CONTRIBUTIONS

We summarize our contributions as follows:

- We propose a novel, sample-efficient algorithm, CAESAR , for the multiple-policy evaluation problem which achieves a non-asymptotic, instance-dependent sample complexity of $\tilde{O}\left(\frac{H^4}{\epsilon^2}\sum_{h=1}^{H}\max_{k\in[K]}\sum_{s,a}\frac{(d_h^{\pi^k}(s,a))^2}{\mu_h^*(s,a)}\right)$. A detailed discussion of this sample complexity, along with a comparison to existing results, is provided in Section 5.
- We introduce the technique of coarse estimation and demonstrate its effectiveness in solving the multiple-policy evaluation problem. We believe this technique has potential applications beyond the scope of this work.
- We propose two algorithms, MARCH and IDES , as subroutines of CAESAR , both of which may be of independent interest. MARCH provides a coarse estimation of the visitation distribution for all deterministic policies, with a sample complexity of $\tilde{O}(\frac{\text{poly}(H,S,A)}{\epsilon})$, despite the exponential number of deterministic policies. IDES offers an accurate estimation of the marginal importance ratio by minimizing a carefully designed step-wise loss function using stochastic gradient descent.
- We introduce a novel metric, termed $\beta$-distance, for the analysis of the MARCH algorithm. We believe that $\beta$-distance may prove valuable in the design of efficient exploration algorithms.

## 4 MAIN RESULTS AND ALGORITHM

In this section, we introduce CAESAR which is sketched out in Algorithm 2 and present the main results. Different from on-policy evaluation, we build a single sampling distribution with which

we can estimate the performance of all target policies using importance weighting. To that end, we first coarsely estimate the visitation distributions of all deterministic policies at the cost of a lower-order sample complexity. Based on these coarse distribution estimators, we can build an optimal sampling distribution by solving a convex optimization problem. Finally, we minimize a carefully designed step-wise loss function using stochastic gradient descent to accurately estimate the importance-weighting ratio. In the following sections, we explain the steps of CAESAR in detail.

## 4.1 COARSE ESTIMATION OF VISITATION DISTRIBUTIONS

We first introduce a proposition that shows how we can coarsely estimate the visitation distributions of target policies with lower-order sample complexity $\tilde{O}(\frac{1}{\epsilon})$. Although this estimator is coarse and cannot be used to directly evaluate the performance of policies, which is our ultimate goal, it possesses nice properties that enable us to construct the optimal sampling distribution and estimate the importance weighting ratio in the following sections.

The idea behind the feasibility of computing these estimators is based on the following lemma which shows that estimating the mean value of a Bernoulli random variable up to constant multiplicative accuracy only requires $\tilde{O}(\frac{1}{\epsilon})$ samples.

**Lemma 4.1.** *Let $Z_\ell$ be i.i.d. samples $Z_\ell \overset{i.i.d.}{\sim} \mathrm{Ber}(p)$. Setting $t \geq \frac{C\log(C/\epsilon\delta)}{\epsilon}$, for some known constant $C > 0$, it follows that with probability at least $1 - \delta$, the empirical mean estimator $\hat{p}_t = \frac{1}{t}\sum_{\ell=1}^t Z_\ell$ satisfies, $|\hat{p}_t - p| \leq \max\{\epsilon, \frac{p}{4}\}$.*

Lemma 4.1 can be used to derive coarse estimators of any policy $\hat{d}^\pi = \{\hat{d}_h^\pi\}_{h=1}^H$, with constant multiplicative accuracy with respect to the true visitation probabilities $d^\pi = \{d_h^\pi\}_{h=1}^H$.

**Proposition 4.2.** *With number of trajectories $n \geq \frac{CK\log(CK/\epsilon\delta)}{\epsilon} = \tilde{O}(\frac{1}{\epsilon})$, we can estimate $\hat{d}^{\pi^k} = \{\hat{d}_h^{\pi^k}\}_{h=1}^H$ such that with probability at least $1-\delta$, $|\hat{d}_h^{\pi^k}(s,a) - d_h^{\pi^k}(s,a)| \leq \max\{\epsilon, \frac{d_h^{\pi^k}(s,a)}{4}\}$, $\forall s \in \mathcal{S}, a \in \mathcal{A}, h \in [H], k \in [K]$.*

Proposition 4.2 works by running each target policy independently and applying Lemma 4.1. However, this would induce an exponential dependency on $S, A$, if for example we aim to coarsely estimate all deterministic policies. To fix this we propose an algorithm named MARCH (see Appendix A.2) that leverages the overlapping visitations of the policy set. Through a novel analysis, we show that MARCH achieves coarse estimation of all deterministic policies with sample complexity $\tilde{O}(\frac{\mathrm{poly}(H,S,A)}{\epsilon})$.

We next show that based on these coarse visitation estimators, we can ignore those states and actions with low estimated visitation probability without inducing significant errors.

**Lemma 4.3.** *Suppose we have an estimator $\hat{d}(s,a)$ of $d(s,a)$ such that $|\hat{d}(s,a) - d(s,a)| \leq \max\{\epsilon', \frac{d(s,a)}{4}\}$. If $\hat{d}(s,a) \geq 5\epsilon'$, then $\max\{\epsilon', \frac{d(s,a)}{4}\} = \frac{d(s,a)}{4}$, and if $\hat{d}(s,a) \leq 5\epsilon'$, then $d(s,a) \leq 7\epsilon'$.*

Based on Lemma 4.3, we can ignore the state-action pairs satisfying $\hat{d}(s,a) \leq 5\epsilon'$. Since if we replace $\epsilon'$ by $\frac{\epsilon}{14SA}$, the error of performance estimation induced by ignoring these state-action pair is at most $\frac{\epsilon}{2}$. For simplicity of presentation, we can set $\hat{d}^\pi(s,a) = d^\pi(s,a) = 0$ if $\hat{d}^\pi(s,a) < \frac{5\epsilon}{14SA}$. Hence, we have that,

$$|\hat{d}_h^{\pi^k}(s,a) - d_h^{\pi^k}(s,a)| \leq \frac{d_h^{\pi^k}(s,a)}{4}, \; \forall s \in \mathcal{S}, a \in \mathcal{A}, h \in [H], k \in [K]. \tag{3}$$

## 4.2 OPTIMAL SAMPLING DISTRIBUTION

We evaluate the expected total rewards of target policies by importance weighting, using samples $\{s_1^i, a_1^i, s_2^i, a_2^i, \ldots, s_H^i, a_H^i\}_{i=1}^n$ drawn from a sampling distribution $\{\mu_h\}_{h=1}^H$. Specifically, $\hat{V}_1^{\pi^k} = \frac{1}{n}\sum_{i=1}^n \sum_{h=1}^H \frac{d_h^{\pi^k}(s_h^i, a_h^i)}{\mu_h(s_h^i, a_h^i)} r_h(s_h^i, a_h^i)$, $k \in [K]$. To minimize the variance of our estimator (see Appendix A.1.2), we find the optimal sampling distribution by solving the following convex

---

**Algorithm 1** **I**mportance **D**ensity **Es**timation (IDES )

---

**Input:** Horizon $H$, accuracy $\epsilon$, target policy $\pi$, coarse estimator $\{\hat{d}_h^\pi\}_{h=1}^H$, $\{\hat{\mu}_h\}_{h=1}^H$, feasible sets $\{D_h\}_{h=1}^H$ where $D_h(s,a) = [0, 2\hat{d}_h^\pi(s,a)]$ and dataset $\mu$.
Initialize $w_h^0 = 0$, $h = 1, \ldots, H$, and set $\mu_0(s_0, a_0) = 1$, $P_0(s|s_0, a_0) = \nu(s)$, $\hat{w}_0 = \hat{\mu}_0 = 1$.
**for** $h = 1$ **to** $H$ **do**

   Set the iteration number of optimization, $n_h = C_h \left( \frac{H^4}{\epsilon^2} \sum_{s,a} \frac{(\hat{d}_h^\pi(s,a))^2}{\hat{\mu}_h(s,a)} + \frac{(\hat{d}_{h-1}^\pi(s,a))^2}{\hat{\mu}_{h-1}(s,a)} \right)$, where

   $C_h$ is a known constant.
   **for** $i = 1$ **to** $n_h$ **do**
     Sample $\{s_h^i, a_h^i\}$ from $\mu_h$ and $\{s_{h-1}^i, a_{h-1}^i, s_h^{i'}\}$ from $\mu_{h-1}$.
     Calculate gradient $g(w_h^{i-1})$,

$$g(w_h^{i-1})(s,a) = \frac{w_h^{i-1}(s,a)}{\hat{\mu}_h(s,a)}\mathbb{I}(s_h^i = s, a_h^i = a) - \frac{\hat{w}_{h-1}(s_{h-1}^i, a_{h-1}^i)}{\hat{\mu}_{h-1}(s_{h-1}^i, a_{h-1}^i)}\pi(a|s)\mathbb{I}(s_h^{i'} = s).$$

     Update $w_h^i = Proj_{w \in D_h}\{w_h^{i-1} - \eta_h^i g(w_h^{i-1})\}$.
   **end for**
   Output the estimator $\hat{w}_h = \frac{1}{\sum_{i=1}^{n_h} 1} \sum_{i=1}^{n_h} w_h^i$.
**end for**

---

optimization problem,

$$\mu_h^* = \arg\min_\mu \max_{k \in [K]} \sum_{s,a} \frac{(d_h^{\pi^k}(s,a))^2}{\mu(s,a)}, \; h \in [H]. \tag{4}$$

However, in some cases, the optimal $\mu^*$ may not be realized by any policy (see Appendix A.1.3). Therefore, to facilitate the construction of the sampling distribution $\mu^*$, we constrain $\mu_h$ to lie within the convex hull of $\mathcal{D} = \{d_h^\pi : \pi \in \Pi_{det}\}$ which formulates the constrained optimization problem,

$$\mu_h^* = \arg\min_{\mu \in \text{conv}(\mathcal{D})} \max_{k \in [K]} \sum_{s,a} \frac{(d_h^{\pi^k}(s,a))^2}{\mu(s,a)}, \; h \in [H]. \tag{5}$$

We denote the optimal solution to (5) as $\mu_h^* = \sum_{\pi \in \Pi_{det}} \alpha_\pi^* d_h^\pi$. Since $d_h^{\pi^k}$ is unknown, we can only solve the approximate optimization problem,

$$\hat{\mu}_h^* = \arg\min_{\mu \in \text{conv}(\hat{\mathcal{D}})} \max_{k \in [K]} \sum_{s,a} \frac{(\hat{d}_h^{\pi^k}(s,a))^2}{\mu(s,a)}, \; h \in [H], \tag{6}$$

where $\hat{\mathcal{D}} = \{\hat{d}_h^\pi : \pi \in \Pi_{det}\}$. We denote the optimal solution to (6) by $\hat{\mu}_h^* = \sum_{\pi \in \Pi_{det}} \hat{\alpha}_\pi^* \hat{d}_h^\pi$. Correspondingly, our real sampling distribution would be $\tilde{\mu}_h^* = \sum_{\pi \in \Pi_{det}} \hat{\alpha}_\pi^* d_h^\pi$.

**Remark 4.1.** *Here, we assume an oracle which gives us the optimal solution of any convex optimization problem. In this work, we focus on the sample complexity which aligns with most theoretical works on reinforcement learning (Amortila et al., 2024a; Liu et al., 2023). We leave it as an open problem of devising a both sample-efficient and computationally efficient algorithm.*

The next lemma tells us that the optimal sampling distribution also has the same property as the coarse distribution estimators.

**Lemma 4.4.** *If property (3) holds:* $|\hat{d}_h^{\pi^k}(s,a) - d_h^{\pi^k}(s,a)| \leq \frac{d_h^{\pi^k}(s,a)}{4}$, $\forall s \in \mathcal{S}, a \in \mathcal{A}, h \in [H], k \in [K]$, then $|\tilde{\mu}_h^*(s,a) - \hat{\mu}_h^*(s,a)| \leq \frac{\tilde{\mu}_h^*(s,a)}{4}$.

### 4.3 ESTIMATION OF THE IMPORTANCE DENSITY

In this section, we introduce the IDES algorithm for estimating the importance weighting ratios which is sketched out in Algorithm 1. IDES is based on the idea of DualDICE Nachum et al. (2019).

In DualDICE, they propose the following loss function

$$\ell^\pi(w) = \frac{1}{2}\mathbb{E}_{s,a\sim\mu}\left[w^2(s,a)\right] - \mathbb{E}_{s,a\sim d^\pi}\left[w(s,a)\right]. \tag{7}$$

The minimum of $\ell^\pi(\cdot)$ is achieved at $w^{\pi,*}(s,a) = \frac{d^\pi(s,a)}{\mu(s,a)}$, the importance weighting ratio. We want to emphasize that IDES is different from DualDICE in many aspects instead of as a simple extension. Specifically, first, IDES employs coarse distribution estimators to tackle the on-policy limitation of the second term in (7), while DualDICE transforms the variable based on Bellman's equation which only works for infinite horizon MDPs. Second, IDES uses a step-wise objective function, requiring step-to-step optimization and analysis, while DualDICE formulates a single loss function. Third, although both IDES and DualDICE achieve a sample complexity of $\tilde{O}(C/\epsilon^2)$, the value of $C$ in DualDICE's bound is not sufficiently tight for our purposes, which involve deriving instance-dependent guarantees. In contrast, we offer a precise analysis linking the value of $C$ in IDES to the expected visitation ratios. Lastly, IDES provides high-probability results for visitation ratio estimation, whereas DualDICE's results hold only in expectation.

More precisely, we define the step-wise loss function of policy $\pi$ at each step $h$ as,

$$\ell_h^\pi(w) = \frac{1}{2}\mathbb{E}_{s,a\sim\tilde{\mu}_h}\left[\frac{w^2(s,a)}{\hat{\mu}_h(s,a)}\right] - \mathbb{E}_{s',a'\sim\tilde{\mu}_{h-1},s\sim P_{h-1}(\cdot|s',a')}\left[\sum_a \frac{\hat{w}_{h-1}(s',a')}{\hat{\mu}_{h-1}(s',a')}w(s,a)\pi(a|s)\right]$$

where $\tilde{\mu}_h = \sum_{\pi\in\Pi_{det}}\hat{\alpha}_\pi^* d_h^\pi$ is the sampling distribution, and $\hat{\mu}_h = \sum_{\pi\in\Pi_{det}}\hat{\alpha}_\pi^* \hat{d}_h^\pi$ is the optimal solution to the approximate optimization problem (6), and we set $\tilde{\mu}_0(s_0,a_0) = 1, P_0(s|s_0,a_0) = \nu(s), \hat{w}_0 = \hat{\mu}_0 = 1$ for notation simplicity.

This loss function possesses two nice properties. First, it is $\gamma$-strongly convex and $\xi$-smooth where $\gamma = \min_{s,a}\frac{\tilde{\mu}_h(s,a)}{\hat{\mu}_h(s,a)}, \xi = \max_{s,a}\frac{\tilde{\mu}_h(s,a)}{\hat{\mu}_h(s,a)}$. Based on the property of our coarse distribution estimator, i.e., $\frac{4}{5} \leq \frac{\tilde{\mu}_h(s,a)}{\hat{\mu}_h(s,a)} \leq \frac{4}{3}$, which is a trivial corollary from Lemma 4.4, $\gamma$ and $\xi$ are bounded as well as their ratio, i.e. $\frac{\xi}{\gamma} \leq \frac{5}{3}$. This property actually plays an important role in deriving the final sample complexity, which we discuss in Appendix A.1.5 due to space constraints.

In the following lemma, we show that our step-wise loss function has friendly step-to-step error propagation properties.

**Lemma 4.5.** *Suppose we have an estimator $\hat{w}_{h-1}$ at step $h-1$ such that,*

$$\sum_{s,a}\left|\tilde{\mu}_{h-1}(s,a)\frac{\hat{w}_{h-1}(s,a)}{\hat{\mu}_{h-1}(s,a)} - d_{h-1}^\pi(s,a)\right| \leq \epsilon,$$

*then by minimizing the loss function $\ell_h^\pi(w)$ at step $h$ to $\|\nabla\ell_h^\pi(\hat{w}_h(s,a))\|_1 \leq \epsilon$, we have,*

$$\sum_{s,a}\left|\tilde{\mu}_h(s,a)\frac{\hat{w}_h(s,a)}{\hat{\mu}_h(s,a)} - d_h^\pi(s,a)\right| \leq 2\epsilon.$$

Lemma 4.5 indicates that using the distribution ratio estimator from the previous step allows us to estimate the ratio at the current step, introducing only an additive error. Consequently, by optimizing step-by-step, we can achieve an accurate estimation of the distribution ratios at all steps, as formalized in the following lemma.

**Lemma 4.6.** *Implementing Algorithm 1, we have the importance density estimator $\frac{\hat{w}_h(s,a)}{\hat{\mu}_h(s,a)}$ such that,*

$$\mathbb{E}\left[\sum_{s,a}\left|\tilde{\mu}_h(s,a)\frac{\hat{w}_h(s,a)}{\hat{\mu}_h(s,a)} - d_h^{\pi^k}(s,a)\right|\right] \leq \frac{\epsilon}{4H}, \ h \in [H]. \tag{8}$$

## 4.4 MAIN RESULTS

We are now ready to present our main sample complexity result for multiple-policy evaluation, building on the results from previous sections. First, we introduce a Median-of-Means (MoM) estimator (Minsker, 2023), formalized in the following lemma, and a data splitting technique that together can be used to convert (8) into a high-probability result (see Appendix A.1.7).

---

**Algorithm 2** **C**oarse and **A**daptive **ES**timation with **A**pproximate **R**eweighing for Multi-Policy Evaluation (CAESAR )

---

**Input:** Accuracy $\epsilon$, confidence $\delta$, target policies $\{\pi^k\}_{k=1}^K$.

Coarsely estimate visitation distributions of all deterministic policies and get $\{\hat{d}^\pi : \pi \in \Pi_{det}\}$.

Solve the approximate optimization problem (6) and obtain $\{\hat{\alpha}_\pi^* : \pi \in \Pi_{det}\}$.

Implement Algorithm 1 with data splitting and obtain MoM estimators $\{\hat{w}^{\pi^k}\}_{k=1}^K$.

Build the final performance estimator $\{\hat{V}_1^{\pi^k}\}_{k=1}^K$ by (9).

**Output:** $\{\hat{V}_1^{\pi^k}\}_{k=1}^K$.

---

**Lemma 4.7.** *Let $x \in \mathbb{R}$ and suppose we have a stochastic estimator $\hat{x}$ such that $\mathbb{E}[|\hat{x} - x|] \leq \frac{\epsilon}{4}$. Then, if we generate $N = O\left(\log(1/\delta)\right)$ i.i.d. estimators $\{\hat{x}_1, \hat{x}_2, \ldots, \hat{x}_N\}$ and choose $\hat{x}_{MoM} = Median(\hat{x}_1, \hat{x}_2, \ldots, \hat{x}_N)$, we have with probability at least $1 - \delta$,*

$$|\hat{x}_{MoM} - x| \leq \epsilon.$$

With the importance density estimator $\frac{\hat{w}_h(s,a)}{\hat{\mu}_h(s,a)}$, we can estimate the performance of policy $\pi^k$,

$$\hat{V}_1^{\pi^k} = \frac{1}{n} \sum_{i=1}^n \sum_{h=1}^H \frac{\hat{w}_h^{\pi^k}(s_h^i, a_h^i)}{\hat{\mu}_h(s_h^i, a_h^i)} r_h(s_h^i, a_h^i). \tag{9}$$

where $\{s_h^i, a_h^i\}_{i=1}^n$ is sampled from $\tilde{\mu}_h$.

We present our main result in the following theorem and leave its detailed derivation to Appendix A.1.7.

**Theorem 4.8.** *Implement Algorithm 2. Then, with probability at least $1 - \delta$, for all target policies, we have that $|\hat{V}_1^{\pi^k} - V_1^{\pi^k}| \leq \epsilon$. And the total number of trajectories sampled is,*

$$n = \tilde{O}\left( \frac{H^4}{\epsilon^2} \sum_{h=1}^H \max_{k \in [K]} \sum_{s,a} \frac{(d_h^{\pi^k}(s,a))^2}{\mu_h^*(s,a)} \right). \tag{10}$$

*Besides, the unknown true visitation distributions can be replaced by the coarse estimator to provide a concrete sample complexity.*

## 5 DISCUSSION

In this section, we analyze our sample complexity, comparing it with existing results and offering several noteworthy findings.

### 5.1 LOWER BOUND AND SOME SPECIAL CASES

For off-policy evaluation, the CR-lower bound proposed by Jiang & Li (2016) (Theorem 3) demonstrates that there exists an MDP such that the variance of any unbiased estimator is lower bounded by $\sum_{h=1}^H \mathbb{E}_\mu \left[ \left( \frac{d_h^\pi(s_h, a_h)}{\mu_h(s_h, a_h)} \right)^2 \mathbb{V}[V_h^\pi(s_h)] \right]$, where $\pi$ is the policy to evaluate and $\mu$ is the sampling distribution. Applying this result to multiple-policy evaluation problem gives us the lower bound[1] $\min_\mu \max_{k \in [K]} \sum_{h=1}^H \mathbb{E}_\mu \left[ \left( \frac{d_h^{\pi^k}(s_h, a_h)}{\mu_h(s_h, a_h)} \right)^2 \mathbb{V}[V_h^{\pi^k}(s_h)] \right]$. From the variance-unaware perspective, where the variance of the value function is simply bounded by $H^2$, our sample complexity matches this lower bound since our sampling distribution is optimal (up to the dependency on $H$). And we believe that a more refined variance-dependent result is achievable and leave it to future works.

---

[1]Here, we discuss the lower bound limited to the method based on a single sampling distribution. A more general lower bound for the multiple-policy evaluation problem that bypasses this single sampling distribution assumption is still an interesting open question.

We then analyze the sample complexity in specific cases, which yield some notable results. First, in the scenario where all target policies are identical, we have $d^{\pi^k} = d, \forall k \in [K]$. In this case, the optimal sampling distribution is $\mu^* = d$, and the sample complexity becomes $\tilde{O}(\frac{H^5}{\epsilon^2})$, showing no dependency on $S$ or $A$ which is consistent with the result from the Monte Carlo sampling method up to the dependency on $H$.

Besides, building on our instance-dependent results, we can derive an instance-independent upper bound for the multiple-policy evaluation problem. Let the sampling distribution $\mu'_h$ be $\frac{1}{SA} \sum_{s,a} d_h^{\pi_{s,a}}$, where $\pi_{s,a} = \arg \max_{k \in [K]} d_h^{\pi^k}(s, a)$. Since $\mu_h^*$ is the optimal solution and $\mu'_h$ is a feasible solution, we have,

$$\max_{k \in [K]} \sum_{s,a} \frac{(d_h^{\pi^k}(s,a))^2}{\mu_h^*(s,a)} \le \max_{k \in [K]} \sum_{s,a} \frac{(d_h^{\pi^k}(s,a))^2}{\mu'_h(s,a)}.$$

By the definition of $\mu'_h$, we have $\sum_{s,a} \frac{(d_h^{\pi^k}(s,a))^2}{\mu'_h(s,a)} \le SA$ which demonstrates that our sample complexity (10) is consistently upper-bounded by $\tilde{O}\left(\frac{H^5 SA}{\epsilon^2}\right)$. Notice that there is a hidden logarithm term $\log(K/\delta)$ in the bound where $K$ is number of policies we aim to evaluate. In the case where we are tasked to evaluate all deterministic policies, $K$ equals $A^S$, leading to an upper bound of $\tilde{O}(\frac{H^5 S^2 A}{\epsilon^2})$. This allows us to identify $\epsilon-$optimal policies for any reward function, effectively solving the reward-free exploration problem (Jin et al., 2020). Our result matches their upper bound and also aligns with the lower bound up to the dependency on $H$.

## 5.2 Comparison with existing results

First, compared to the naive uniform sampling strategy over target policies as described in (2), our method has a clear advantage. Our sampling distribution is optimal among all possible combinations of the target policies, including the naive uniform strategy.

Next, we compare our result with the one achieved by Dann et al. (2023) as described in (1). A significant issue with the result by Dann et al. (2023) is the presence of the unfavorable $\frac{1}{d^{max}(s)}$, which can induce an undesirable dependency on $K$.

To illustrate this, consider an example of an MDP with two layers: a single initial state $s_{1,1}$ in the first layer and two terminal states in the second layer $s_{2,1}, s_{2,2}$. The transition function is the same for all actions, i.e., $P(s_{2,1}|s_{1,1}, a) = p$ and $p$ is sufficiently small. Agents only receive rewards at state $s_{2,1}$, regardless of the actions they take. Hence, to evaluate the performance of a policy under this MDP, it is sufficient to consider only the second layer. Now, suppose we have $K$ target policies to evaluate, where each policy takes different actions at state $s_{1,1}$ but the same action at any state in the second layer. Since the transition function at state $s_{1,1}$ is the same for any action, the visitation distribution at state $s_{2,1}$ of all target policies is identical. Given that $p$ is sufficiently small, the probability of reaching $s_{2,1}$ is $\mathbb{P}[s_{2,1} \in \mathcal{K}^2] = 1 - (1-p)^K \approx pK$. According to the result (1) by Dann et al. (2023), the sample complexity in this scenario is $\tilde{O}(\frac{K}{\epsilon^2})$ which depends on $K$. In contrast, since the visitation distribution at the second layer of all target policies is identical, our result provides a sample complexity of $\tilde{O}(\frac{1}{\epsilon^2})$ without dependency on $K$. Nevertheless, it remains unclear whether our result is universally better in all cases (omit the dependency on $H$).

## 5.3 Near-optimal policy identification

Besides policy evaluation, CAESAR can also be applied to identify a near-optimal policy. Fixing the high-probability factor, we denote the sample complexity of CAESAR by $\tilde{O}(\frac{\Theta(\Pi)}{\gamma^2})$, where $\Pi$ is the set of policies to be evaluated and $\gamma$ is the estimation error. We provide a simple algorithm based on CAESAR in Appendix A.4 that achieves an instance-dependent sample complexity $\tilde{O}(\max_{\gamma \ge \epsilon} \frac{\Theta(\Pi_\gamma)}{\gamma^2})$ to identify a $\epsilon-$optimal policy, where $\Pi_\gamma = \{\pi : V_1^* - V_1^\pi \le 8\gamma\}$. This result is interesting as it offers a different perspective beyond the existing gap-dependent results (Simchowitz & Jamieson, 2019; Dann et al., 2021). Furthermore, this result can be easily extended to the multi-reward setting. Due to space constraints, we leave the detailed discussion to Appendix A.4.

## 6 CONCLUSION AND FUTURE WORK

In this work, we consider the problem of multi-policy evaluation. We propose an algorithm, CAESAR, based on computing an approximate optimal offline sampling distribution and using the data sampled from it to perform the simultaneous estimation of the policy values. CAESAR uses $n = \tilde{O}\left(\frac{H^4}{\epsilon^2}\sum_{h=1}^{H}\max_{k\in[K]}\sum_{s,a}\frac{(d_h^{\pi^k}(s,a))^2}{\mu_h^*(s,a)}\right)$ trajectories and with probability at least $1-\delta$ we can evaluate the performance of all target policies up to an $\epsilon$ error. The algorithm consists of three techniques. First, we obtain a coarse distribution estimator at the cost of lower-order sample complexity. Second, based on the coarse distribution estimator, we show an achievable optimal sampling distribution by solving a convex optimization problem. Last, we propose a novel step-wise loss function for finite-horizon MDPs. By minimizing the loss function step to step, we are able to get the importance weighting ratio and a non-asymptotic sample complexity is available due to the smoothness and strong-convexity of the loss function.

Beyond the results of this work, there are still some open questions of interest. First, our sample complexity has a dependency on $H^4$ which is induced by the error propagation in the estimation of the importance weighting ratios. Specifically, the error of minimizing the loss function at early steps, e.g., $h = 1$ will propagate to later steps e.g., $h = H$. We conjecture a dependency on $H^2$ is possible by considering a comprehensive loss function which includes the entire horizon instead of step-wise loss functions which require step by step optimization. Second, as discussed before, we believe that a variance-aware sample complexity is possible through a more careful analysis. Besides, considering a reward-dependent sample complexity is also an interesting direction. For example, in an MDP with sparse rewards where only one state-action has non-zero reward; a better sample complexity than CAESAR 's may be possible by focusing on state-action pairs with non-zero rewards. Finally, we are interested to see what other uses the research community may find for coarse distribution estimation. In our work, the coarse distribution estimator plays an important role throughout the algorithm. We believe this type of estimator can be of independent interest.

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

## A  APPENDIX

### A.1  PROOF OF THEOREMS AND LEMMAS IN SECTION 4

#### A.1.1  PROOF OF LEMMA 4.1

Our results relies on the following variant of Bernstein inequality for martingales, or Freedman's inequality (Freedman, 1975), as stated in e.g., (Agarwal et al., 2014; Beygelzimer et al., 2011).

**Lemma A.1** (Simplified Freedman's inequality). *Let* $X_1, ..., X_T$ *be a bounded martingale difference sequence with* $|X_\ell| \leq R$. *For any* $\delta' \in (0, 1)$, *and* $\eta \in (0, 1/R)$, *with probability at least* $1 - \delta'$,

$$\sum_{\ell=1}^{T} X_\ell \leq \eta \sum_{\ell=1}^{T} \mathbb{E}_\ell[X_\ell^2] + \frac{\log(1/\delta')}{\eta}. \tag{11}$$

*where* $\mathbb{E}_\ell[\cdot]$ *is the conditional expectation[2] induced by conditioning on* $X_1, \cdots, X_{\ell-1}$.

**Lemma A.2** (Anytime Freedman). *Let* $\{X_t\}_{t=1}^{\infty}$ *be a bounded martingale difference sequence with* $|X_t| \leq R$ *for all* $t \in \mathbb{N}$. *For any* $\delta' \in (0, 1)$, *and* $\eta \in (0, 1/R)$, *there exists a universal constant* $C > 0$ *such that for all* $t \in \mathbb{N}$ *simultaneously with probability at least* $1 - \delta'$,

$$\sum_{\ell=1}^{t} X_\ell \leq \eta \sum_{\ell=1}^{t} \mathbb{E}_\ell[X_\ell^2] + \frac{C \log(t/\delta')}{\eta}. \tag{12}$$

*where* $\mathbb{E}_\ell[\cdot]$ *is the conditional expectation induced by conditioning on* $X_1, \cdots, X_{\ell-1}$.

*Proof.* This result follows from Lemma A.1. Fix a time-index $t$ and define $\delta_t = \frac{\delta'}{12t^2}$. Lemma A.1 implies that with probability at least $1 - \delta_t$,

$$\sum_{\ell=1}^{t} X_\ell \leq \eta \sum_{\ell=1}^{t} \mathbb{E}_\ell\left[X_\ell^2\right] + \frac{\log(1/\delta_t)}{\eta}.$$

A union bound implies that with probability at least $1 - \sum_{\ell=1}^{t} \delta_t \geq 1 - \delta'$,

$$\sum_{\ell=1}^{t} X_\ell \leq \eta \sum_{\ell=1}^{t} \mathbb{E}_\ell\left[X_\ell^2\right] + \frac{\log(12t^2/\delta')}{\eta}$$

$$\overset{(i)}{\leq} \eta \sum_{\ell=1}^{t} \mathbb{E}_\ell\left[X_\ell^2\right] + \frac{C \log(t/\delta')}{\eta}.$$

holds for all $t \in \mathbb{N}$. Inequality $(i)$ holds because $\log(12t^2/\delta') = \mathcal{O}\left(\log(t\delta')\right)$.

$\square$

**Proposition A.3.** *Let* $\delta' \in (0, 1)$, $\beta \in (0, 1]$ *and* $Z_1, \cdots, Z_T$ *be an adapted sequence satisfying* $0 \leq Z_\ell \leq \tilde{B}$ *for all* $\ell \in \mathbb{N}$. *There is a universal constant* $C' > 0$ *such that,*

$$(1 - \beta) \sum_{t=1}^{T} \mathbb{E}_t[Z_t] - \frac{2\tilde{B}C' \log(T/\delta')}{\beta} \leq \sum_{\ell=1}^{T} Z_\ell \leq (1 + \beta) \sum_{t=1}^{T} \mathbb{E}_t[Z_t] + \frac{2\tilde{B}C' \log(T/\delta')}{\beta}$$

*with probability at least* $1 - 2\delta'$ *simultaneously for all* $T \in \mathbb{N}$.

*Proof.* Consider the martingale difference sequence $X_t = Z_t - \mathbb{E}_t[Z_t]$. Notice that $|X_t| \leq \tilde{B}$. Using the inequality of Lemma A.2 we obtain that for all $\eta \in (0, 1/B^2)$

$$\sum_{\ell=1}^{t} X_\ell \leq \eta \sum_{\ell=1}^{t} \mathbb{E}_\ell[X_\ell^2] + \frac{C \log(t/\delta')}{\eta}$$

$$\overset{(i)}{\leq} 2\eta B^2 \sum_{\ell=1}^{t} \mathbb{E}_\ell[Z_\ell] + \frac{C \log(t/\delta')}{\eta},$$

---

[2]We will use this notation to denote conditional expectations throughout this work.

for all $t \in \mathbb{N}$ with probability at least $1 - \delta'$. Inequality $(i)$ holds because $\mathbb{E}_t[X_t^2] \leq B^2\mathbb{E}[|X_t|] \leq 2B^2\mathbb{E}_t[Z_t]$ for all $t \in \mathbb{N}$. Setting $\eta = \frac{\beta}{2B^2}$ and substituting $\sum_{\ell=1}^t X_\ell = \sum_{\ell=1}^t Z_\ell - \mathbb{E}_\ell[Z_\ell]$,

$$\sum_{\ell=1}^t Z_\ell \leq (1 + \beta) \sum_{\ell=1}^t \mathbb{E}_\ell[Z_\ell] + \frac{2B^2 C \log(t/\delta')}{\beta} \tag{13}$$

with probability at least $1 - \delta'$. Now consider the martingale difference sequence $X_t' = \mathbb{E}[Z_t] - Z_t$ and notice that $|X_t'| \leq B^2$. Using the inequality of Lemma A.2 we obtain for all $\eta \in (0, 1/B^2)$,

$$\sum_{\ell=1}^t X_\ell' \leq \eta \sum_{\ell=1}^t \mathbb{E}_\ell[(X_\ell')^2] + \frac{C \log(t/\delta')}{\eta}$$

$$\leq 2\eta B^2 \sum_{\ell=1}^t \mathbb{E}_\ell[Z_\ell] + \frac{C \log(t/\delta')}{\eta}.$$

Setting $\eta = \frac{\beta}{2B^2}$ and substituting $\sum_{\ell=1}^t X_\ell' = \sum_{\ell=1}^t \mathbb{E}[Z_\ell] - Z_\ell$ we have,

$$(1 - \beta) \sum_{\ell=1}^t \mathbb{E}[Z_\ell] \leq \sum_{\ell=1}^t Z_\ell + \frac{2B^2 C \log(t/\delta')}{\beta} \tag{14}$$

with probability at least $1 - \delta'$. Combining Equations 13 and 14 and using a union bound yields the desired result.

$$\square$$

Let the $Z_\ell$ be i.i.d. samples $Z_\ell \overset{i.i.d.}{\sim} \mathrm{Ber}(p)$. The empirical mean estimator, $\widehat{p}_t = \frac{1}{t} \sum_{\ell=1}^t Z_\ell$ satisfies,

$$(1 - \beta)p - \frac{2C' \log(t/\delta')}{\beta t} \leq \widehat{p}_t \leq (1 + \beta)p + \frac{2C' \log(t/\delta')}{\beta t}$$

with probability at least $1 - 2\delta'$ for all $t \in \mathbb{N}$ where $C' > 0$ is a (known) universal constant. Given $\epsilon > 0$ set $t \geq \frac{8C' \log(t/\delta')}{\beta \epsilon}$ (notice the dependence of $t$ on the RHS - this can be achieved by setting $t \geq \frac{C \log(C/\beta\epsilon\delta')}{\beta \epsilon}$ for some (known) universal constant $C > 0$).

In this case observe that,

$$(1 - \beta)p - \epsilon/8 \leq \widehat{p}_t \leq (1 + \beta)p + \epsilon/8.$$

Setting $\beta = 1/8$,

$$7p/8 - \epsilon/8 \leq \widehat{p}_t \leq 9p/8 + \epsilon/8,$$

so that,

$$p - \widehat{p}_t \leq p/8 + \epsilon/8,$$

and

$$\widehat{p}_t - p \leq p/8 + \epsilon/8,$$

which implies $|\widehat{p}_t - p| \leq p/8 + \epsilon/8 \leq 2 \max(p/8, \epsilon/8) = \max(p/4, \epsilon/4)$.

### A.1.2 DERIVATION OF THE OPTIMAL SAMPLING DISTRIBUTION (4)

Our performance estimator is,

$$\hat{V}_1^{\pi^k} = \frac{1}{n} \sum_{i=1}^n \sum_{h=1}^H \frac{d_h^{\pi^k}(s_h^i, a_h^i)}{\mu_h(s_h^i, a_h^i)} r(s_h^i, a_h^i), \ k \in [K].$$

Denote $\sum_{h=1}^{H} \frac{d_h^{\pi^k}(s_h^i, a_h^i)}{\mu_h(s_h^i, a_h^i)} r_h(s_h^i, a_h^i)$ by $X_i$. And for simplicity, denote $\mathbb{E}_{(s_1,a_1)\sim\mu_1,\ldots,(s_H,a_H)\sim\mu_H}$ by $\mathbb{E}_\mu$, the variance of our estimator is bounded by,

$$\mathbb{E}_\mu[X_i^2] = \mathbb{E}_\mu \left[ \left( \sum_{h=1}^{H} \frac{d_h^{\pi^k}(s_h^i, a_h^i)}{\mu_h(s_h^i, a_h^i)} r_h(s_h^i, a_h^i) \right)^2 \right]$$

$$\leq \mathbb{E}_\mu \left[ H \cdot \sum_{h=1}^{H} \left( \frac{d_h^{\pi^k}(s_h^i, a_h^i)}{\mu_h(s_h^i, a_h^i)} r_h(s_h^i, a_h^i) \right)^2 \right]$$

$$\leq \mathbb{E}_\mu \left[ H \cdot \sum_{h=1}^{H} \left( \frac{d_h^{\pi^k}(s_h^i, a_h^i)}{\mu_h(s_h^i, a_h^i)} \right)^2 \right]$$

$$= H \cdot \sum_{h=1}^{H} \mathbb{E}_{d_h^{\pi^k}} \left[ \frac{d_h^{\pi^k}(s_h^i, a_h^i)}{\mu_h(s_h^i, a_h^i)} \right].$$

The first inequality holds by Cauchy $-$ Schwarz inequality. The second inequality holds due to the assumption $r_h(s, a) \in [0, 1]$.

Denote $\sum_{h=1}^{H} \mathbb{E}_{d_h^{\pi^k}} \left[ \frac{d_h^{\pi^k}(s_h^i, a_h^i)}{\mu_h(s_h^i, a_h^i)} \right]$ by $\rho_{\mu,k}$. Applying Bernstein's inequality, we have that with probability at least $1 - \delta$ and $n$ samples, it holds,

$$|\hat{V}_1^{\pi^k} - V_1^{\pi^k}| \leq \sqrt{\frac{2H\rho_{\mu,k}\log(1/\delta)}{n}} + \frac{2M_k\log(1/\delta)}{3n},$$

where $M_k = \max_{s_1,a_1,\ldots,s_H,a_H} \sum_{h=1}^{H} \frac{d_h^{\pi^k}(s_h,a_h)}{\mu_h(s_h,a_h)} r_h(s_h, a_h)$.

To achieve an $\epsilon$ accuracy of evaluation, we need samples,

$$n_{\mu,k} \leq \frac{8H\rho_{\mu,k}\log(1/\delta)}{\epsilon^2} + \frac{4M_k\log(1/\delta)}{3\epsilon}.$$

Take the union bound over all target policies,

$$n_\mu \leq \frac{8H\max_{k\in[K]}\rho_{\mu,k}\log(K/\delta)}{\epsilon^2} + \frac{4M\log(K/\delta)}{3\epsilon},$$

where $M = \max_{k\in[K]} M_k$.

We define the optimal sampling distribution $\mu^*$ as the one minimizing the higher order sample complexity,

$$\mu_h^* = \arg\min_{\mu_h} \max_{k\in[K]} \mathbb{E}_{d_h^{\pi^k}(s,a)} \left[ \frac{d_h^{\pi^k}(s,a)}{\mu_h(s,a)} \right]$$

$$= \arg\min_{\mu_h} \max_{k\in[K]} \sum_{s,a} \frac{\left( d_h^{\pi^k}(s,a) \right)^2}{\mu_h(s,a)}, \quad h = 1, \ldots, H.$$

### A.1.3 AN EXAMPLE OF UNREALIZABLE OPTIMAL SAMPLING DISTRIBUTION

Here, we give an example to illustrate the assertation that in some cases, the optimal sampling distribution cannot be realized by any policy.

Consider such a MDP with two layers, in the first layer, there is a single initial state $s_{1,1}$, in the second layer, there are two states $s_{2,1}, s_{2,2}$. The transition function at state $s_{1,1}$ is identical for any action, $\mathbb{P}(s_{2,1}|s_{1,1}, a) = \mathbb{P}(s_{2,2}|s_{1,1}, a) = \frac{1}{2}$. Hence, for any policy, the only realizable state visitation distribution at the second layer is $d_2(s_{2,1}) = d_2(s_{2,2}) = \frac{1}{2}$.

Suppose the target policies take $K \geq 2$ different actions at state $s_{2,1}$ while take the same action at state $s_{2,2}$.

By solving the optimization problem (4), we have the optimal sampling distribution at the second layer,

$$\mu_2^*(s_{2,1}) = \frac{K^2}{1+K^2}, \ \mu_2^*(s_{2,2}) = \frac{1}{1+K^2},$$

which is clearly not realizable by any policy.

### A.1.4  PROOF OF LEMMA 4.5

*Proof.* The gradient of $\ell_h^\pi(w)$ is,

$$\nabla_{w(s,a)}\ell_h^\pi(w) = \frac{\tilde{\mu}_h(s,a)}{\hat{\mu}_h(s,a)}w(s,a) - \sum_{s',a'}\tilde{\mu}_{h-1}(s',a')P(s|s',a')\pi(a|s)\frac{\hat{w}_{h-1}(s',a')}{\hat{\mu}_{h-1}(s',a')}.$$

Suppose by some SGD algorithm, we can converge to a point $\hat{w}_h$ such that the gradient of the loss function is less than $\epsilon$,

$$\|\nabla\ell_h^\pi(\hat{w}_h)\|_1 = \sum_{s,a}\left|\frac{\tilde{\mu}_h(s,a)}{\hat{\mu}_h(s,a)}\hat{w}_h(s,a) - \sum_{s',a'}\tilde{\mu}_{h-1}(s',a')P(s|s',a')\pi(a|s)\frac{\hat{w}_{h-1}(s',a')}{\hat{\mu}_{h-1}(s',a')}\right| \le \epsilon.$$

By decomposing,

$$\left|\frac{\tilde{\mu}_h(s,a)}{\hat{\mu}_h(s,a)}\hat{w}_h(s,a) - \sum_{s',a'}\tilde{\mu}_{h-1}(s',a')P(s|s',a')\pi(a|s)\frac{\hat{w}_{h-1}(s',a')}{\hat{\mu}_{h-1}(s',a')}\right|$$

$$= \left|\frac{\tilde{\mu}_h(s,a)}{\hat{\mu}_h(s,a)}\hat{w}_h(s,a) - d_h^\pi(s,a) + d_h^\pi(s,a) - \sum_{s',a'}\tilde{\mu}_{h-1}(s',a')P(s|s',a')\pi(a|s)\frac{\hat{w}_{h-1}(s',a')}{\hat{\mu}_{h-1}(s',a')}\right|$$

$$\ge \left|\frac{\tilde{\mu}_h(s,a)}{\hat{\mu}_h(s,a)}\hat{w}_h(s,a) - d_h^\pi(s,a)\right| - \left|d_h^\pi(s,a) - \sum_{s',a'}\tilde{\mu}_{h-1}(s',a')P(s|s',a')\pi(a|s)\frac{\hat{w}_{h-1}(s',a')}{\hat{\mu}_{h-1}(s',a')}\right|$$

$$= \left|\tilde{\mu}_h(s,a)\frac{\hat{w}_h(s,a)}{\hat{\mu}_h(s,a)} - d_h^\pi(s,a)\right|$$

$$- \left|\sum_{s',a'}P(s|s',a')\pi(a|s)\left(d_{h-1}^\pi(s',a') - \tilde{\mu}_{h-1}(s',a')\frac{\hat{w}_{h-1}(s',a')}{\hat{\mu}_{h-1}(s',a')}\right)\right|.$$

Hence, we have,

$$\sum_{s,a}\left|\tilde{\mu}_h(s,a)\frac{\hat{w}_h(s,a)}{\hat{\mu}_h(s,a)} - d_h^\pi(s,a)\right|$$

$$\le \epsilon + \sum_{s,a}\left|\sum_{s',a'}P(s|s',a')\pi(a|s)\left(d_{h-1}^\pi(s',a') - \tilde{\mu}_{h-1}(s',a')\frac{\hat{w}_{h-1}(s',a')}{\hat{\mu}_{h-1}(s',a')}\right)\right|$$

$$\le \epsilon + \sum_{s',a'}\left|d_{h-1}^\pi(s',a') - \tilde{\mu}_{h-1}(s',a')\frac{\hat{w}_{h-1}(s',a')}{\hat{\mu}_{h-1}(s',a')}\right|$$

$$\le 2\epsilon.$$

$\square$

### A.1.5  PROOF OF LEMMA 4.6

*Proof.* The minimum $w_h^*$ of the loss function $\ell_h^\pi(w)$ is $w_h^*(s,a) = \frac{d_h^\pi(s,a)}{\tilde{\mu}_h(s,a)}\hat{\mu}_h(s,a)$ if $\hat{w}_{h-1}$ achieves optimum. By the property of the coarse distribution estimator, we have,

$$w_h^*(s,a) = \frac{d_h^\pi(s,a)}{\tilde{\mu}_h(s,a)}\hat{\mu}_h(s,a) \le \frac{\frac{4}{3}\hat{d}_h^\pi(s,a)}{\frac{4}{5}\hat{\mu}_h(s,a)}\hat{\mu}_h(s,a) = \frac{5}{3}\hat{d}_h^\pi(s,a).$$

We can define a feasible set for the optimization problem, i.e. $w_h(s,a) \in [0, D_h(s,a)], D_h(s,a) = 2\hat{d}_h^\pi(s,a)$.

Next, we analyse the variance of the stochastic gradient. We denote the stochastic gradient as $g_h(w)$, $\{s_1^i, a_1^i, \ldots, s_H^i, a_H^i\}$ a trajectory sampled from $\tilde{\mu}_h$ and $\{s_1^j, a_1^j, \ldots, s_H^j, a_H^j\}$ a trajectory sampled from $\tilde{\mu}_{h-1}$.

$$g_h(w)(s,a) = \frac{w(s,a)}{\hat{\mu}_h(s,a)}\mathbb{I}(s_h^i = s, a_h^i = a) - \frac{\hat{w}_{h-1}(s_{h-1}^j, a_{h-1}^j)}{\hat{\mu}_{h-1}(s_{h-1}^j, a_{h-1}^j)}\pi(a|s)\mathbb{I}(s_h^j = s).$$

The variance bound becomes

$$\mathbb{V}[g_h(w)] \leq \mathbb{E}[\|g_h(w)\|^2] \leq \sum_{s,a} \tilde{\mu}_h(s,a)\left(\frac{w(s,a)}{\hat{\mu}_h(s,a)}\right)^2 + \tilde{\mu}_{h-1}(s,a)\left(\frac{\hat{w}_{h-1}(s,a)}{\hat{\mu}_{h-1}(s,a)}\right)^2$$

$$\leq O\left(\sum_{s,a} \frac{(\hat{d}_h^\pi(s,a))^2}{\hat{\mu}_h(s,a)} + \frac{(\hat{d}_{h-1}^\pi(s,a))^2}{\hat{\mu}_{h-1}(s,a)}\right), \tag{15}$$

where the last inequality is due to the bounded feasible set for $w$ and the property of coarse distribution estimator $\tilde{\mu}_h(s,a) \leq \frac{4}{3}\hat{\mu}_h(s,a)$.

Based on the error propagation lemma 4.5, if we can achieve $\|\nabla\ell_h^\pi(\hat{w}_h)\|_1 \leq \frac{\epsilon}{4H^2}$ from step $h = 1$ to step $h = H$, then we have,

$$\sum_{s,a}\left|\tilde{\mu}_h(s,a)\frac{\hat{w}_h(s,a)}{\hat{\mu}_h(s,a)} - d_h^\pi(s,a)\right| \leq \frac{\epsilon}{4H}, \forall h = 1, 2, \ldots, H,$$

which can enable us to build the final estimator of the performance of policy $\pi$ with at most error $\epsilon$.

By the property of smoothness, to achieve $\|\nabla\ell_h^\pi(\hat{w}_h)\|_1 \leq \frac{\epsilon}{4H^2}$, we need to achieve $\ell_h^\pi(\hat{w}_h) - \ell_h^\pi(w_h^*) \leq \frac{\epsilon^2}{32\xi H^4}$ where $\xi$ is the smoothness factor, because,

$$\|\nabla\ell_h^\pi(\hat{w}_h)\|_1^2 \leq 2\xi(\ell_h^\pi(\hat{w}_h) - \ell_h^\pi(w_h^*)) \leq \frac{\epsilon^2}{16H^4}.$$

**Lemma A.4.** *For a $\lambda-$strongly convex loss function $L(w)$ satisfying $\|w^*\| \leq D$ for some known $D$, there exists a stochastic gradient descent algorithm that can output $\hat{w}$ after $T$ iterations such that,*

$$\mathbb{E}[L(\hat{w}) - L(w^*)] \leq \frac{2G^2}{\lambda(T+1)},$$

*where $G^2$ is the variance bound of the stochastic gradient.*

Invoke the convergence rate for strongly-convex and smooth loss functions, i.e. Lemma A.4, we have that the number of samples needed to achieve $\ell_h^\pi(\hat{w}_h) - \ell_h^\pi(w_h^*) \leq \frac{\epsilon^2}{32\xi H^4}$ is,

$$n = O\left(\frac{\xi}{\gamma}\frac{H^4 G^2}{\epsilon^2}\right).$$

We have shown in Section 4.3 that $\frac{\xi}{\gamma} \leq \frac{5}{3}$, this nice property helps us to get rid of the undesired ratio of the smoothness factor and the strongly-convexity factor, i.e. $\frac{\max_{s,a}\mu(s,a)}{\min_{s,a}\mu(s,a)}$ of the original loss function (7) which can be extremely bad. Replacing $G^2$ by our variance bound (15), we have,

$$n_h^\pi = O\left(\frac{H^4}{\epsilon^2}\left(\sum_{s,a}\frac{(\hat{d}_h^\pi(s,a))^2}{\hat{\mu}_h(s,a)} + \frac{(\hat{d}_{h-1}^\pi(s,a))^2}{\hat{\mu}_{h-1}(s,a)}\right)\right).$$

For each step $h$, we need the above number of trajectories, sum over $h$, we have the total sample complexity,

$$n^\pi = O\left(\frac{H^4}{\epsilon^2}\sum_{h=1}^{H}\sum_{s,a}\frac{(\hat{d}_h^\pi(s,a))^2}{\hat{\mu}_h(s,a)}\right).$$

To evaluate $K$ policies, we need trajectories,

$$n = O\left(\frac{H^4}{\epsilon^2} \sum_{h=1}^{H} \max_{k \in [K]} \sum_{s,a} \frac{(\hat{d}_h^{\pi^k}(s,a))^2}{\hat{\mu}_h(s,a)}\right).$$

$\square$

### A.1.6 PROOF OF LEMMA 4.7

*Proof.* By Markov's inequality, we have,

$$\mathbb{P}(|\hat{\mu} - \mu| \geq \epsilon) \leq \frac{\mathbb{E}[|\hat{\mu} - \mu|]}{\epsilon} \leq \frac{1}{4}.$$

The event that $|\hat{\mu}_{MoM} - \mu| > \epsilon$ belongs to the event where more than half estimators $\hat{\mu}_i$ are outside of the desired range $|\hat{\mu}_i - \mu| > \epsilon$, hence, we have,

$$\mathbb{P}(|\hat{\mu}_{MoM} - \mu| > \epsilon) \leq \mathbb{P}(\sum_{i=1}^{N} \mathbb{I}(|\hat{\mu}_i - \mu| > \epsilon) \geq \frac{N}{2}).$$

Denote $\mathbb{I}(|\hat{\mu}_i - \mu| > \epsilon)$ by $Z_i$ and $\mathbb{E}[Z_i] = p$,

$$\mathbb{P}(|\hat{\mu}_{MoM} - \mu| > \epsilon) = \mathbb{P}(\sum_{i=1}^{N} Z_i \geq \frac{N}{2})$$

$$= \mathbb{P}(\frac{1}{N} \sum_{i=1}^{N} (Z_i - p) \geq \frac{1}{2} - p)$$

$$\leq e^{-2N(\frac{1}{2}-p)^2}$$

$$\leq e^{-\frac{N}{8}},$$

where the first inequality holds by Hoeffding's inequality and the second inequality holds due to $p \leq \frac{1}{4}$. Set $\delta = e^{-\frac{N}{8}}$, we have, with $N = O(\log(1/\delta))$, with probability at least $1 - \delta$, it holds $|\hat{\mu}_{MoM} - \mu| \leq \epsilon$. $\square$

### A.1.7 PROOF OF THEOREM 4.8

Here, we explain how Theorem 4.8 is derived. We first show how the Median-of-Means (MoM) estimator and data splitting technique can conveniently convert Lemma 4.6 to a version holds with high probability.

For step $h$, Algorithm 1 can output a solution $\hat{w}_h$ such that $\mathbb{E}[\ell_h^\pi(\hat{w}_h) - \ell_h^\pi(w_h^*)] \leq \frac{\epsilon^2}{32\xi H^4}$. We can apply Lemma 4.7 on our algorithm which means that we can run the algorithm for $N = O(\log(1/\delta))$ times. Hence, we will get $N$ solutions $\{\hat{w}_{h,1}, \hat{w}_{h,2}, \ldots, \hat{w}_{h,N}\}$. Set $\hat{w}_{h,MoM}$ as the solution such that $\ell_h^\pi(\hat{w}_{h,MoM}) = \text{Median}(\ell_h^\pi(\hat{w}_{h,1}), \ell_h^\pi(\hat{w}_{h,2}), \ldots, \ell_h^\pi(\hat{w}_{h,N}))$. Based on Lemma 4.7, we have that with probability at least $1 - \delta$, it holds $\ell_h^\pi(\hat{w}_{h,MoM}) - \ell_h^\pi(w_h^*) \leq \frac{\epsilon^2}{32\xi H^4}$. With a little abuse of notation, we just denote $\hat{w}_{h,MoM}$ by $\hat{w}_h$ in the following content.

Now we are ready to estimate the total expected rewards of target policies, With the importance weighting ratio estimator $\frac{\hat{w}_h(s,a)}{\hat{\mu}_h(s,a)}$ from Algorithm 1, we can estimate the performance of policy $\pi^k$,

$$\hat{V}_1^{\pi^k} = \frac{1}{n} \sum_{i=1}^{n} \sum_{h=1}^{H} \frac{\hat{w}_h^{\pi^k}(s_h^i, a_h^i)}{\hat{\mu}_h(s_h^i, a_h^i)} r_h(s_h^i, a_h^i), \tag{16}$$

where $\{s_h^i, a_h^i\}_{i=1}^{n}$ is sampled from $\tilde{\mu}_h$.

**Lemma A.5.** *With samples* $n = \tilde{O}\left(\frac{H^2}{\epsilon^2} \sum_{h=1}^{H} \max_{k \in [K]} \sum_{s,a} \frac{(\hat{d}_h^{\pi^k}(s,a))^2}{\hat{\mu}_h(s,a)}\right)$, *we have with proba-bility at least* $1 - \delta$, $|\hat{V}_1^{\pi^k} - V_1^{\pi^k}| \leq \frac{\epsilon}{2}$, $k \in [K]$.

*Proof.* First, we can decompose the error $|\hat{V}_1^{\pi^k} - V_1^{\pi^k}| = |\hat{V}_1^{\pi^k} - \mathbb{E}[\hat{V}_1^{\pi^k}] + \mathbb{E}[\hat{V}_1^{\pi^k}] - V_1^{\pi^k}| \leq$ $|\hat{V}_1^{\pi^k} - \mathbb{E}[\hat{V}_1^{\pi^k}]| + |\mathbb{E}[\hat{V}_1^{\pi^k}] - V_1^{\pi^k}|$. Then, by Bernstein's inequality, with samples $n = \tilde{O}\left(\frac{H^2}{\epsilon^2}\sum_{h=1}^H \max_{k\in[K]}\sum_{s,a}\frac{(\hat{d}_h^{\pi^k}(s,a))^2}{\hat{\mu}_h(s,a)}\right)$, we have, $|\hat{V}_1^{\pi^k} - \mathbb{E}[\hat{V}_1^{\pi^k}]| \leq \frac{\epsilon}{4}$. Based Lemma 4.6, we have, $|\mathbb{E}[\hat{V}_1^{\pi^k}] - V_1^{\pi^k}| \leq \frac{\epsilon}{4}$. $\qquad\square$

Remember that in Section 4.1, we ignore those states and actions with low estimated visitation distribution for each target policy which induce at most $\frac{\epsilon}{2}$ error. Combined with Lemma A.5, our estimator $\hat{V}_1^{\pi^k}$ finally achieves that with probability at least $1 - \delta$, $|\hat{V}_1^{\pi^k} - V_1^{\pi^k}| \leq \epsilon, k \in [K]$.

And for sample complexity, in our algorithm, we need to sample data in three procedures. First, for the coarse estimation of the visitation distribution, we need $\tilde{O}(\frac{1}{\epsilon})$ samples. Second, to estimate the importance-weighting ratio, we need samples $\tilde{O}\left(\frac{H^4}{\epsilon^2}\sum_{h=1}^H \max_{k\in[K]}\sum_{s,a}\frac{(d_h^{\pi^k}(s,a))^2}{\mu_h^*(s,a)}\right)$. Last, to build the final performance estimator (9), we need samples $\tilde{O}\left(\frac{H^2}{\epsilon^2}\sum_{h=1}^H \max_{k\in[K]}\sum_{s,a}\frac{(\hat{d}_h^{\pi^k}(s,a))^2}{\hat{\mu}_h(s,a)}\right)$. Therefore, the total trajectories needed,

$$n = \tilde{O}\left(\frac{H^4}{\epsilon^2}\sum_{h=1}^H \max_{k\in[K]}\sum_{s,a}\frac{(d_h^{\pi^k}(s,a))^2}{\mu_h^*(s,a)}\right).$$

Moreover, notice that,

$$\max_{k\in[K]}\sum_{s,a}\frac{(\hat{d}_h^{\pi^k}(s,a))^2}{\hat{\mu}_h(s,a)} \leq \max_{k\in[K]}\sum_{s,a}\frac{(\hat{d}_h^{\pi^k}(s,a))^2}{\mu_h^*(s,a)} \leq \frac{25}{16}\sum_{s,a}\frac{(d_h^{\pi}(s,a))^2}{\mu_h^*(s,a)}, \qquad (17)$$

where $\mu_h^*$ is the optimal solution of the optimization problem (5), the first inequality holds due to $\hat{\mu}_h$ is the minimum of the approximate optimization problem (6) and the second inequality holds due to $\hat{d}_h^{\pi}(s,a) \leq \frac{5}{4}d_h^{\pi}(s,a)$. Based on (17), we can substitute the coarse distribution estimator in the sample complexity bound by the exact one,

$$n = \tilde{O}\left(\frac{H^4}{\epsilon^2}\sum_{h=1}^H \max_{k\in[K]}\sum_{s,a}\frac{(d_h^{\pi^k}(s,a))^2}{\mu_h^*(s,a)}\right).$$

## A.2 LOWER ORDER COARSE ESTIMATION

---

**Algorithm 3** **M**ulti-policy **A**pproximation via **R**atio-based **C**oarse **H**andling (MARCH)

---

**Input:** Horizon $H$, accuracy $\epsilon$, policy $\pi$.
Coarsely estimate $d_1$ such that $dist^\beta(\hat{d}_1, d_1) \leq \epsilon$, where $\beta = \frac{1}{H}$.
**for** $h = 1$ **to** $H - 1$ **do**
   1. Coarsely estimate $\mu_h$ such that $|\hat{\mu}_h(s,a) - \mu_h(s,a)| \leq \max\{\epsilon', c \cdot \mu_h(s,a)\}$, where $\epsilon' = \frac{\epsilon}{2H^2S^2A^2}$ and $c = \frac{\beta}{2}$.
   2. Sample $\{s_h^i, a_h^i, s_{h+1}^i\}_{i=1}^n$ from $\mu_h$.
   3. Estimate $d_{h+1}(s,a)$ by $\hat{d}_{h+1}(s,a) = \frac{1}{n}\sum_{i=1}^n \mathbb{I}(s_{h+1}^i = s)\hat{w}_h(s_h^i, a_h^i)$.
**end for**
**Output:** $\{\hat{d}_h\}_{h=1}^H$.

---

In this section, we first provide our algorithm MARCH for coarse estimation of all the deterministic policies and then conduct an analysis on its sample complexity.

MARCH is based on the algorithm EULER proposed by Zanette & Brunskill (2019).

**Lemma A.6** (Theorem 3.3 in Jin et al. (2020))**.** *Based on* EULER*, with sample complexity* $\tilde{O}(\frac{poly(H,S,A)}{\epsilon})$*, we can construct a policy cover which generates a dataset with the distribution* $\mu$ *such that, with probability* $1 - \delta$*, if* $d_h^{max}(s) \geq \frac{\epsilon}{SA}$*, then,*

$$\mu_h(s,a) \geq \frac{d_h^{max}(s,a)}{2HSA}, \tag{18}$$

*where* $d_h^{max}(s) = \max_\pi d_h^\pi(s), d_h^{max}(s,a) = \max_\pi d_h^\pi(s,a)$*.*

With this dataset, we estimate the visitation distribution of deterministic policies by step-to-step importance weighting,

$$\hat{d}_{h+1}(s,a) = \frac{1}{n}\sum_{i=1}^n \mathbb{I}(s_{h+1}^i = s)\hat{w}_h(s_h^i, a_h^i),$$

where $\{s_h^i, a_h^i, s_{h+1}^i\}_{i=1}^n$ are sampled from $\mu$ and $\hat{w}_h(s,a) = \frac{\hat{d}_h(s,a)}{\hat{\mu}_h(s,a)}$.

We state that MARCH can coarsely estimate the visitation distributions of all the deterministic policies by just paying a lower-order sample complexity which is formalized in the following theorem.

**Theorem A.7.** *Implement Algorithm 3 with the number of trajectories* $n = \tilde{O}(\frac{poly(H,S,A)}{\epsilon})$*, with probability at least* $1 - \delta$*, it holds that for any deterministic policy* $\pi$*,*

$$|\hat{d}_h^\pi(s,a), d_h^\pi(s,a)| \leq \max\{\epsilon, \frac{d_h^\pi(s,a)}{4}\}, \ \forall s \in \mathcal{S}, a \in \mathcal{A}, h \in [H],$$

*where* $\hat{d}^\pi$ *is the distribution estimator.*

*Proof.* Our analysis is based a notion of distance defined in the following.

**Definition A.1** ($\beta-$distance)**.** *For* $x, y \geq 0$*, we define the* $\beta-$*distance as,*

$$dist^\beta(x,y) = \min_{\alpha \in [\frac{1}{\beta}, \beta]} |\alpha x - y|.$$

*Correspondingly, for* $x, y \in \mathbb{R}^n$*,*

$$dist^\beta(x,y) = \sum_{i=1}^n dist^\beta(x_i, y_i).$$

Based on its definition, we show in the following lemma that $\beta-$distance has some properties.

**Lemma A.8.** *The $\beta-$distance possesses the following properties for $(x, y, z, \gamma \geq 0)$:*

$$1. \ dist^\beta(\gamma x, \gamma y) = \gamma dist^\beta(x, y); \tag{19}$$

$$2. \ dist^\beta(x_1 + x_2, y_1 + y_2) \leq dist^\beta(x_1, y_1) + dist^\beta(x_2, y_2); \tag{20}$$

$$3. \ dist^{\beta_1 \cdot \beta_2}(x, z) \leq dist^{\beta_1}(x, y) \cdot \beta_2 + dist^{\beta_2}(y, z). \tag{21}$$

*Proof.* See Appendix A.3.1. □

The following lemma shows that if we can control the $\beta-$distance between $\hat{x}, x$, then we can show $\hat{x}$ achieves the coarse estimation of $x$.

**Lemma A.9.** *Suppose $dist^{1+\beta}(x, y) \leq \epsilon$, then it holds that,*

$$|x - y| \leq \beta y + (1 + \frac{\beta}{1 + \beta})\epsilon \leq 2\max\{(1 + \frac{\beta}{1 + \beta})\epsilon, \beta y\}.$$

*Proof.* See Appendix A.3.2. □

The logic of the analysis is to show the $\beta-$distance between $\hat{d}_h$ and $d_h$ can be bounded at each layer by induction. Then by Lemma A.9, we show $\{\hat{d}_h\}_{h=1}^H$ achieves coarse estimation.

Suppose at layer $h$, we have $\hat{d}_h$ such that $dist^{(1+\beta)^h}(\hat{d}_h, d_h) < \epsilon_h$ where $\beta = \frac{1}{H}$. For notation simplicity, we omit the superscript $\pi$. The analysis holds for any policy.

We use importance weighting to estimate $\hat{d}_{h+1}$,

$$\hat{d}_{h+1}(s, a) = \frac{1}{n}\sum_{i=1}^n \mathbb{I}(s_{h+1}^i = s)\pi(a|s)\hat{w}_h(s_h^i, a_h^i),$$

where $\hat{w}_h(s, a) = \frac{\hat{d}_h(s,a)}{\hat{\mu}_h(s,a)}$.

We also denote,

$$\overline{d}_{h+1}(s, a) = \mathbb{E}_{(s_h, a_h, s_{h+1})\sim\mu_h}[\mathbb{I}(s_{h+1} = s)\hat{w}_h(s_h, a_h)].$$

By (21) in Lemma A.8, we have,

$$dist^{(1+\beta)^{h+2}}(\hat{d}_{h+1}, d_{h+1}) \leq \underbrace{dist^{(1+\beta)}(\hat{d}_{h+1}, \overline{d}_{h+1})(1+\beta)^{h+1}}_{A} + \underbrace{dist^{(1+\beta)^{h+1}}(\overline{d}_{h+1}, d_{h+1})}_{B}. \tag{22}$$

Next, we show how we can bound these two terms $(A)$ and $(B)$. Note that for $(s, h)$ where $d_h^{max}(s) < \frac{\epsilon}{SA}$, the induced $\beta-$distance error is at most $\epsilon$. Therefore, we can just discuss state-action pairs which satisfy Lemma A.6.

**Bound of** $(A)$   We first show the following lemma tells us that the importance weighting is upper-bounded.

**Lemma A.10.** *Based on the definition of $\mu$, the importance weighting is upper bounded,*

$$w_h(s, a) = \frac{d_h(s,a)}{\mu_h(s,a)} \leq 2HSA\frac{d_h(s,a)}{d_h^{max}(s,a)} \leq 2HSA.$$

*Hence, we can clip $\hat{w}_h(s, a)$ at $2HSA$ such that $\hat{w}_h(s, a) \leq 2HSA$.*

Let's define the random variable $Z_{h+1}(s, a) = \mathbb{I}(s_{h+1} = s)\hat{w}_h(s_h, a_h)$, then $\hat{d}_{h+1}(s, a) = \frac{1}{n}\sum_{i=1}^n Z_{h+1}^i(s, a)$. Since $\hat{w}_h(s_h, a_h)$ is bounded by Lemma A.10, we have,

$$\mathbb{V}[Z_{h+1}(s, a)] \leq \mathbb{E}[Z_{h+1}(s, a)^2] \leq 2HSA\mathbb{E}[Z_{h+1}(s, a)].$$

By Berstein's inequality, we have with probability at least $1 - \delta$,

$$|\hat{d}_{h+1}(s,a) - \mathbb{E}[\hat{d}_{h+1}(s,a)]| \leq \sqrt{\frac{2\mathbb{V}[Z_{h+1}(s,a)]\log(1/\delta)}{n}} + \frac{2HSA\log(1/\delta)}{3n}$$

$$\leq \sqrt{\frac{4HSA\mathbb{E}[\hat{d}_{h+1}(s,a)]\log(1/\delta)}{n}} + \frac{2HSA\log(1/\delta)}{3n},$$

to achieve the estimation accuracy $|\hat{d}_{h+1}(s,a) - \mathbb{E}[\hat{d}_{h+1}(s,a)]| \leq \max\{\epsilon, c \cdot \mathbb{E}[\hat{d}_{h+1}(s,a)]\}$, we need samples $n = \tilde{O}\left(\frac{HSA}{c \cdot \epsilon}\right)$.

Based on the above analysis, we can achieve,

$$|\hat{d}_{h+1}(s,a), \overline{d}_{h+1}(s,a)| \leq \max\{\epsilon', \frac{\beta}{2}\overline{d}_{h+1}(s,a)\}$$

at the cost of samples $\tilde{O}\left(\frac{HSA}{\beta\epsilon'}\right)$.

We now show $dist^{1+\beta}(\hat{d}_{h+1}, \overline{d}_{h+1}) \leq SA\epsilon'$. We discuss it in two cases,

$$1. \ |\hat{d}_{h+1}(s,a), \overline{d}_{h+1}(s,a)| \leq \epsilon' \tag{23}$$

$$2. \ |\hat{d}_{h+1}(s,a), \overline{d}_{h+1}(s,a)| \leq \frac{\beta}{2}\overline{d}_{h+1}(s,a). \tag{24}$$

For those $(s,a)$ which satisfies (24), since $[1 - \frac{\beta}{2}, 1 + \frac{\beta}{2}] \in [\frac{1}{1+\beta}, 1 + \beta]$, by the definition of $\beta-$distance, we have,

$$dist^{1+\beta}(\hat{d}_{h+1}(s,a), \overline{d}_{h+1}(s,a)) = 0. \tag{25}$$

For other $(s,a)$ which satisfies (23), we have,

$$dist^{1+\beta}(\hat{d}_{h+1}(s,a), \overline{d}_{h+1}(s,a)) \leq |\hat{d}_{h+1}(s,a), \overline{d}_{h+1}(s,a)| \leq \epsilon'.$$

Since there are at most $SA$ state-action pairs, the error in the second case is at most $SA\epsilon'$. Combine these two cases, we have,

$$dist^{1+\beta}(\hat{d}_{h+1}, \overline{d}_{h+1}) \leq SA\epsilon'.$$

By setting $\epsilon = \frac{\epsilon'}{SA}$, we have,

$$(A) = dist^{1+\beta}(\hat{d}_{h+1}, \overline{d}_{h+1})(1 + \beta)^{h+1} \leq (1 + \beta)^{h+1}\epsilon, \tag{26}$$

and the sample complexity is $\tilde{O}\left(\frac{(HSA)^2}{\epsilon}\right)$.

**Bound of** $(B)$    Next we show how to bound term $(B)$. Denote $\mu_h(s,a)\frac{\hat{d}_h(s,a)}{\hat{\mu}_h(s,a)}$ by $\tilde{d}_h(s,a)$, we have,

$$(B) = dist^{(1+\beta)^{h+1}}(\overline{d}_{h+1}, d_{h+1})$$

$$= \sum_{s,a} dist^{(1+\beta)^{h+1}}(\overline{d}_{h+1}(s,a), d_{h+1}(s,a))$$

$$= \sum_{s,a} dist^{(1+\beta)^{h+1}}(\sum_{s',a'} P_h^\pi(s,a|s',a')\tilde{d}_h(s',a'), \sum_{s',a'} P_h^\pi(s,a|s',a')d_h(s',a'))$$

$$\leq \sum_{s,a}\sum_{s',a'} dist^{(1+\beta)^{h+1}}(P_h^\pi(s,a|s',a')\tilde{d}_h(s',a'), P_h^\pi(s,a|s',a')d_h(s',a'))$$

$$= \sum_{s,a}\sum_{s',a'} P_h^\pi(s,a|s',a')dist^{(1+\beta)^{h+1}}(\tilde{d}_h(s',a'), d_h(s',a'))$$

$$= dist^{(1+\beta)^{h+1}}(\tilde{d}_h, d_h),$$

where the first two equality holds by definition, the inequality holds by (20) in Lemma A.8, the third equality holds by (19) in Lemma A.8 and the last one holds by $\sum_{s,a} P_h^\pi(s,a|s',a') = 1$.

Now we analyse $dist^{(1+\beta)^{h+1}}(\tilde{d}_h, d_h)$.

$$dist^{(1+\beta)^{h+1}}(\tilde{d}_h, d_h) = \sum_{s,a} \mu_h(s,a) dist^{(1+\beta)^{h+1}}(\frac{\hat{d}_h(s,a)}{\hat{\mu}_h(s,a)}, \frac{d_h(s,a)}{\mu_h(s,a)}).$$

By coarse estimation, we have $|\hat{\mu}_h(s,a) - \mu_h(s,a)| \leq \max\{\epsilon', c \cdot \mu_h(s,a)\}$. Similarly, we discuss it in two cases,

$$1.\ |\hat{\mu}_h(s,a), \mu_h(s,a)| \leq \epsilon', \tag{27}$$

$$2.\ |\hat{\mu}_h(s,a), \mu_h(s,a)| \leq c \cdot \mu_h(s,a). \tag{28}$$

For those $(s,a)$ which satisfies (27), by Lemma A.10, we have,

$$dist^{(1+\beta)^{h+1}}(\frac{\hat{d}_h(s,a)}{\hat{\mu}_h(s,a)}, \frac{d_h(s,a)}{\mu_h(s,a)}) \leq |\frac{\hat{d}_h(s,a)}{\hat{\mu}_h(s,a)} - \frac{d_h(s,a)}{\mu_h(s,a)}| \leq 2HSA.$$

Hence, we have,

$$dist^{(1+\beta)^{h+1}}(\tilde{d}_h(s,a), d_h(s,a)) = \mu_h(s,a) dist^{(1+\beta)^{h+1}}(\frac{\hat{d}_h(s,a)}{\hat{\mu}_h(s,a)}, \frac{d_h(s,a)}{\mu_h(s,a)})$$

$$\leq 2HSA\mu_h(s,a) \leq \frac{2HSA\epsilon'}{c},$$

where the last inequality holds by $c \cdot \mu_h(s,a) \leq \epsilon'$.

Next, For those $(s,a)$ which satisfies (28), we have,

$$(1-c)\frac{1}{\hat{\mu}_h(s,a)} \leq \frac{1}{\mu_h(s,a)} \leq (1+c)\frac{1}{\hat{\mu}_h(s,a)}.$$

Set $c = \frac{\beta}{2}$, since $[1 - \frac{\beta}{2}, 1 + \frac{\beta}{2}] \in [\frac{1}{1+\beta}, 1+\beta]$, by definition of $\beta-$distance, we have,

$$dist^{(1+\beta)}(\frac{1}{\hat{\mu}_h(s,a)}, \frac{1}{\mu_h(s,a)}) = 0. \tag{29}$$

And we assume by induction that $dist^{(1+\beta)^h}(\hat{d}_h(s,a), d_h(s,a)) \leq \epsilon_h$, together with (29) we have,

$$dist^{(1+\beta)^{h+1}}(\frac{\hat{d}_h(s,a)}{\hat{\mu}_h(s,a)}, \frac{d_h(s,a)}{\mu_h(s,a)}) \leq \epsilon_h. \tag{30}$$

Combine the results of two cases together, we have,

$$(B) = dist^{(1+\beta)^{h+1}}(\tilde{d}_h, d_h) \leq \epsilon_h + 4H^2S^2A^2\epsilon'$$

Set $\epsilon' = \frac{\epsilon}{4H^2S^2A^2}$, we have,

$$(B) \leq \epsilon_h + \epsilon \tag{31}$$

at the cost of samples $\tilde{O}(\frac{H^3S^2A^2}{\epsilon})$.

Now we are ready to show the bound of $\beta-$distance at layer $h+1$. Plug (26)(31) into (22), we have,

$$dist^{(1+\beta)^{h+2}}(\hat{d}_{h+1}, d_{h+1}) \leq dist^{(1+\beta)}(\hat{d}_{h+1}, \overline{d}_{h+1})(1+\beta)^{h+1} + dist^{(1+\beta)^{h+1}}(\overline{d}_{h+1}, d_{h+1})$$

$$\leq (1+\beta)^{h+1}\epsilon + \epsilon + \epsilon_h.$$

Start from $dist^{(1+\beta)}(\hat{d}_1, d_1) \leq \epsilon$, we have,

$$dist^{(1+\beta)^{2h-1}}(\hat{d}_h, d_h) \leq h\epsilon + \epsilon\sum_{l=1}^{h-1}(1+\beta)^{2h}. \tag{32}$$

Remember that $\beta = \frac{1}{H}$ and due to $(1 + \frac{1}{H})^h \leq e$ $(h \leq H)$, we have,

$$dist^{e^2}(\hat{d}_h, d_h) \leq H(1+e^2)\epsilon. \tag{33}$$

Recall Lemma A.9, and based on (33), we have,

$$|\hat{d}_h(s,a) - d_h(s,a)| \leq 2\max\{H(1+e^2)\epsilon, (e^2-1)d_h(s,a)\}.$$

By just paying multiplicative constant, we can adjust the constant above to meet our needs, i.e. in Theorem A.7. $\qquad\square$

### A.3  PROOF OF LEMMAS IN SECTION A.2

#### A.3.1  PROOF OF LEMMA A.8

*Proof.* 1. The first property is trivial.

$$dist^\beta(\gamma x, \gamma y) = \min_{\alpha \in [\frac{1}{\beta}, \beta]} |\alpha \gamma x - \gamma y|$$

$$= \min_{\alpha \in [\frac{1}{\beta}, \beta]} \gamma |\alpha x - y|$$

$$= \gamma dist^\beta(x, y).$$

2. Let $\alpha_i$ be such that,

$$dist^{1+\beta}(x_i, y_i) = |\alpha_i x_i - y_i|, \ i = 1, 2.$$

Notice that $\alpha_3 = \alpha_1 \cdot \frac{x_1}{x_1 + x_2} + \alpha_2 \cdot \frac{x_2}{x_1 + x_2}$ satisfies $\alpha_3 \in [\alpha_1, \alpha_2] \in [\frac{1}{\beta}, \beta]$ and $\alpha_3(x_1 + x_2) = \alpha_1 x_1 + \alpha_2 x_2$, therefore,

$$dist^\beta(x_1 + x_2, y_1 + y_2) = \min_{\alpha \in [\frac{1}{\beta}, \beta]} |\alpha(x_1 + x_2) - y_1 - y_2|$$

$$\leq |\alpha_3(x_1 + x_2) - y_1 - y_2|$$

$$= |\alpha_1 x_1 + \alpha_2 x_2 - y_1 - y_2|$$

$$\leq |\alpha_1 x_1 - y_1| + |\alpha_2 x_2 - y_2|$$

$$= dist^\beta(x_1, y_1) + dist^\beta(x_2, y_2).$$

The first inequality holds due to the definition of $\beta-$distance. The second inequality is the triangle inequality.

3. We prove the third property through a case-by-case discussion.

(1). $\frac{x}{\beta_1 \beta_2} \leq z \leq \beta_1 \beta_2 x$. In this case, the result is trivial, since $dist^{\beta_1 \beta_2}(x, z) = 0$ and $\beta-$distance is always non-negative.

(2). $\beta_1 \beta_2 x < z$. If $y \leq x$, then,

$$dist^{\beta_1 \beta_2}(x, z) \leq dist^{\beta_2}(x, z) \leq dist^{\beta_2}(y, z).$$

We are done.

If $x < y \leq \beta_1 x$, then $dist_1^\beta(x, y) = 0$, and $z > \beta_1 \beta_2 x \geq \beta_2 y$, hence,

$$dist^{\beta_2}(y, z) = z - \beta_2 y \geq z - \beta_1 \beta_2 x = dist^{\beta_1 \beta_2}(x, z).$$

We are done.

If $y > \beta_1 x, z \in [\frac{y}{\beta_2}, \beta_2 y]$, then,

$$dist^{\beta_1}(x, y)\beta_2 + dist^{\beta_2}(y, z) = \beta_2(y - \beta_1 x)$$

$$\geq z - \beta_1 \beta_2 x$$

$$= dist^{\beta_1 \beta_2}(x, z).$$

We are done.

If $y > \beta_1 x, z \notin [\frac{y}{\beta_2}, \beta_2 y]$, then,

$$dist^{\beta_1}(x, y)\beta_2 + dist^{\beta_2}(y, z) \geq \beta_2(y - \beta_1 x)$$

$$\geq z - \beta_1 \beta_2 x$$

$$= dist^{\beta_1 \beta_2}(x, z).$$

We are done.

(3). $z < \frac{x}{\beta_1 \beta_2}$. A symmetric analysis can be done by replacing $\beta_1, \beta_2$ by $\frac{1}{\beta_1}, \frac{1}{\beta_2}$ which gives the result,

$$dist^{\beta_1 \beta_2}(x, z) \leq dist^{\beta_1}(x, y)\frac{1}{\beta_2} + dist^{\beta_2}(y, z)$$

Since $\beta_2 \geq 1$ and $dist^{\beta_1}(x,y) \geq 0$, we have $dist^{\beta_1}(x,y)\frac{1}{\beta_2} \leq dist^{\beta_1}(x,y)\beta_2$, hence,

$$dist^{\beta_1\beta_2}(x,z) \leq dist^{\beta_1}(x,y)\beta_2 + dist^{\beta_2}(y,z),$$

which concludes the proof. □

### A.3.2 Proof of Lemma A.9

*Proof.* We prove the lemma through a case-by-case study.

(1). $x \leq y$. If $dist^{1+\beta}(x,y) = 0$, then $x(1+\beta) \geq y \geq x$, therefore,

$$|x - y| = y - x \leq \beta x \leq \beta y.$$

If $dist^{1+\beta}(x,y) > 0$, then $dist^{1+\beta}(x,y) = y - (1+\beta)x$, therefore,

$$|x - y| = y - x = dist^{1+\beta}(x,y) + \beta x \leq \epsilon + \beta x \leq \epsilon + \beta y.$$

(2). $y < x$. If $dist^{1+\beta}(x,y) = 0$, then $\frac{x}{1+\beta} \leq y < x$, therefore,

$$|x - y| = x - y \leq x - \frac{x}{1+\beta} \leq y(1+\beta)(1 - \frac{1}{1+\beta}) = \beta y.$$

If $dist^{1+\beta}(x,y) > 0$, then $y < \frac{x}{1+\beta} \leq x$ and $dist^{1+\beta}(x,y) = \frac{x}{1+\beta} - y$. Moreover, since $dist^{1+\beta}(x,y) \leq \epsilon$, we have $\frac{x}{1+\beta} \leq \epsilon + y$. Therefore,

$$|x - y| = x - y$$
$$= dist^{1+\beta}(x,y) + (1 - \frac{1}{1+\beta})x$$
$$= dist^{1+\beta}(x,y) + \beta\frac{x}{1+\beta}$$
$$\leq \epsilon + \frac{\beta}{1+\beta}\epsilon + \beta y$$
$$= (1 + \frac{\beta}{1+\beta})\epsilon + \beta y.$$

Combine the results above together, we have,

$$|x - y| \leq \beta y + (1 + \frac{\beta}{1+\beta})\epsilon \leq 2\max\{(1 + \frac{\beta}{1+\beta})\epsilon, \beta y\}.$$

□

## A.4   DISCUSSION ON POLICY IDENTIFICATION

In this section, we discuss on the application of CAESAR to policy identification problem, its instance-dependent sample complexity and some intuitions related to the existing gap-dependent results.

We first provide a simple algorithm that utilizes CAESAR to identify an $\epsilon-$optimal policy. The core idea behind the algorithm is we can use CAESAR to evaluate all candidate policies up to an accuracy, then we can eliminate those policies with low estimated performance. By decreasing the evaluation error gradually, we can finally identify a near-optimal policy with high probability.

For notation simplicity, fixing the high-probability factor, we denote the sample complexity of CAESAR by $\frac{\Theta(\Pi)}{\gamma^2}$, where $\Pi$ is the set of policies to be evaluated and $\gamma$ is the estimation error.

---

**Algorithm 4** Policy Identification based on CAESAR

    **Input:** Alg CAESAR , optimal factor $\epsilon$, candidate policy set $\Pi$.
    **for** $i = 1$ **to** $\lceil \log_2(4/\epsilon) \rceil$ **do**
        1. Run CAESAR to evaluate the performance of policies in $\Pi$ up to accuracy $\gamma = \frac{1}{2^i}$.
        2. Eliminate $\pi^i$ if $\exists \pi^j \in \Pi, \hat{V}_1^{\pi^j} - \hat{V}_1^{\pi^i} > 2\gamma$, update $\Pi$.
    **end for**
    **Output:** Randomly pick $\pi^o$ from $\Pi$.

---

**Theorem A.11.** *Implement Algorithm 4, we have that, with probability at least $1 - \delta$, $\pi^o$ is $\epsilon-$optimal, i.e.,*

$$V_1^* - V_1^{\pi^o} \le \epsilon.$$

*And the instance-dependent sample complexity is $\tilde{O}(\max_{\gamma \ge \epsilon} \frac{\Theta(\Pi_\gamma)}{\gamma^2})$, where $\Pi_\gamma = \{\pi : V_1^* - V_1^\pi \le 8\gamma\}$.*

*Proof.* On the one hand, based on the elimination rule in the algorithm, by running CAESAR with the evaluation error $\gamma$, the optimal policy $\pi^*$ will not be eliminated with probability at least $1 - \delta$. Since $\max_{\pi \in \Pi} \hat{V}_1^\pi - \hat{V}_1^{\pi^*} \le V_1^* + \gamma - (V_1^{\pi^*} - \gamma) \le 2\gamma$.

On the other hand, if $V_1^* - V_1^{\pi^i} > 4\gamma$, then $\pi^i$ will be eliminated with probability at least $1 - \delta$. Since $\max_{\pi \in \Pi} \hat{V}_1^\pi - \hat{V}_1^{\pi^i} > V_1^* - \gamma - (V_1^{\pi^i} + \gamma) > 2\gamma$.

Therefore, by running Algorithm 4, the final policy set is not empty and for any policy $\pi$ in this set, it holds, $V_1^* - V_1^\pi \le \epsilon$ with probability at least $1 - \delta$.

Next, we analyse the sample complexity of Algorithm 4. Based on above analysis, within every iteration of the algorithm, we have a policy set containing $8\gamma-$optimal policies, and we use CAESAR to evaluate the performance of these policies up to $\gamma$ accuracy. By Theorem 4.8, the sample complexity is $\frac{\Theta(\Pi_\gamma)}{\gamma^2}$. Therefore, the overall sample complexity is,

$$\sum_\gamma \frac{\Theta(\Pi_\gamma)}{\gamma^2} \le \tilde{O}(\max_{\gamma \ge \epsilon} \frac{\Theta(\Pi_\gamma)}{\gamma^2}).$$

$\square$

This result is quite interesting since it provides another perspective beyond the existing gap-dependent results for policy identification. And these two results have some intuitive relations that may be of interest.

Roughly speaking, to identify an $\epsilon-$optimal policy for an MDP, the gap-dependent regret is described as,

$$O(\sum_{h,s,a} \frac{H \log K}{gap_h(s,a)}),$$

where $gap_h(s,a) = V_h^*(s) - Q_h^*(s,a)$.

The value gap $gap_h(s,a)$ quantifies how sub-optimal the action $a$ is at state $s$. If the gap is small, it is difficult to distinguish and eliminate the sub-optimal action. At the same time, smaller gaps mean that there are more policies with similar performance to the optimal policy, i.e. the policy set $\Pi_\gamma$ is larger. Both our result and gap-dependent result can capture this intuition. We conjecture there exists a quantitative relationship between these two perspectives.

An interesting proposition of Theorem A.11 is to apply the same algorithm to the multi-reward setting. A similar instance-dependent sample complexity can be achieved $\tilde{O}(\max_{\gamma \geq \epsilon} \frac{\Theta(\Pi_\gamma^{\mathcal{R}})}{\gamma^2})$ with the difference that $\Pi_\gamma^{\mathcal{R}}$ contains policies which is $8\gamma-$optimal for at least one reward function. This sample complexity captures the intrinsic difficulty of the problem by how similar the near-optimal policies under different rewards are which is consistent with the intuition.

