# OpenReview forum: "Sample Efficient Multiple-policy Evaluation in Reinforcement Learning"
_ICLR.cc/2025/Conference — Submitted to ICLR 2025_

### Official Review · Reviewer_7jyM · 2024-11-03

**Soundness:** 2
**Presentation:** 3
**Contribution:** 1
**Rating:** 3
**Confidence:** 5

**Summary:**

The paper studies the off-policy evaluation problem in offline settings. The authors introduce an important sampling-based estimation procedure for policy evaluation. In general, the estimation process consists of two parts: the first part is to estimate the visitation of the target polices and in the second part, the density ratio and the offline distribution are estimated via the data. The sample complexity over the performance error is established.

**Strengths:**

1. The paper studies a finite sample guarantee for the performance error.
2. The underlying problem, i.e., off-policy evaluation, is important in offline RL.
3. The structure of the paper is clear.

**Weaknesses:**

1. The paper misses a lot of the important results in the OPE literature:
     a. Uehara, Masatoshi, Jiawei Huang, and Nan Jiang. "Minimax weight and q-function learning for off-policy evaluation." ICML
     b. Kallus, Nathan, et al. "Doubly robust distributionally robust off-policy evaluation and learning." ICML
     c. Liu, Yao, Pierre-Luc Bacon, and Emma Brunskill. "Understanding the curse of horizon in off-policy evaluation via conditional
          importance sampling." ICML
     d. Kallus, Nathan, Yuta Saito, and Masatoshi Uehara. "Optimal off-policy evaluation from multiple logging policies." ICML
The above works (but not limited to) are highly related to the current paper, from the marginalized important sampling estimator to doubly robust estimator, and then include the evaluation of multiple logging policies. The lack of discussions on these works limits the novelty of the current work.

2. The scenarios and applications for the simultaneous OPE on multiple policies are not convincing. OPE for single/mixture/logging policy is very important in RL literature, however, the current framework in this work lacks evidence of the real-life application.

3. In the offline RL setting, the distributional shift is one of the most important issues, in both OPE and OPL problems. Unfortunately, in this work, the study and explicit treatment of this issue does not appear. See the review: Levine, Sergey, et al. "Offline reinforcement learning: Tutorial, review, and perspectives on open problems." arXiv preprint arXiv:2005.01643 (2020).

4. In recent years, there has been significant progress in relaxing the coverage condition in offline RL literature. In many works, the single policy concentrability condition is enough to learn a policy with a sample efficient guarantee in the policy learning setting. Therefore, the full coverage assumption, which is implicitly assumed in this work, is very stringent. This greatly limits the potential application scope in practice.  See the one of the related papers: Jin, Ying, Zhuoran Yang, and Zhaoran Wang. "Is pessimism provably efficient for offline rl?." ICML

5. The calculation of the importance weighting ratios is very difficult in practice. The challenges come from the Realizability and Stabablity. The choices of the function class are not well discussed in this function approximation setting. Also, the practical optimization stability problem is not discussed.

6. Theorem 4.8: The sample complexity in this work is almost not improved in terms of the intuitive $K$-policy extension of the existing work in Uehara, Masatoshi, et al. "Finite sample analysis of minimax offline reinforcement learning: Completeness, fast rates and first-order efficiency." arXiv.

7. One of the major concerns in this work is the lack of empirical evaluation of the benchmark. One cannot position this work in the existing literature without evidence from comprehensive empirical studies. Especially, considering the complexity of the estimation procedure and the optimization stability issue behind it, this practical application of this algorithm is unknown.

**Questions:**

See weakness.

---

> ### Author Response · Authors · 2024-11-17
>
> Thanks for your time and feedback. We kindly request the reviewer to consider increasing the score based on our careful responses and clarifications below.
>
> Based on the summary, we think the reviewer misunderstood the main focus and the contribution of our paper.
> 1. The reviewer concluded in the summary that 'The paper studies the off-policy evaluation problem in offline settings', this is a totally misleading summary of our work. Our work studies the setting where the interaction with the environment is permitted, instead of offline setting.
> 2. Our work studies multi-policy evaluation problem (it is in the title), the difficulty of this problem comes from the fact that we have multiple policies to evaluate and how we can leverage the similarity between these policies to save the samples collected. However, the review didn't talk about multiple-policy at all in the summary which is our main focus.
> 3. The reviewer wrote 'the first part is to estimate the visitation of the target polices and in the second part, the density ratio and the offline distribution are estimated via the data' which is again wrong or ignores the key points. Our main contribution of the first phase of our algorithm is the coarse estimation of all deterministic and target policies. The review just ignores 'coarse estimation'. The sampling distribution is not estimated via data in the second phase. It is calculated by solving an opt problem based on the coarse estimator from the first phase.

---

> > ### Author Response · Authors · 2024-11-17
> >
> > For the weaknesses,
> > > 1. Thanks for listing these papers. We didn't include these papers because they are not very related to the main focus of our work. These papers try to improve the off-policy estimator based on a given offline dataset. Our setting is different where we are not given a dataset, instead we learn it. And we tackle the multiple-policy evaluation problem where the focus is how to leverage the similarity between these policies instead of improving an off-policy estimator. We will consider to include them since they are related in a broad way.
> >
> > >2. As we mentioned in the paper, one example of motivation is to evaluate multiple policies trained using different hyperparameters. Besides, our multiple-policy evaluation algorithm actually can also solve optimal-policy identification problem (see section 5.3) with a similar result to existing works for finding an optimal policy.
> >
> > > 3&4. The reviewer keeps talking about offline RL, again, for clear clarification, our work is not under the offline setting. Hence, the comments in weaknesses 3&4 are not correct. In the first phase of our algorithm, at the cost of low-order samples, by interacting with the environment, we learn a distribution with good coverage from scratch without any coverage assumption.
> >
> > > 5. We are confused at the reviewer's comment on 'The choices of the function class are not well discussed in this function approximation setting' since in our work, we study the tabular setting. The function approximation setting is more general yet more difficult, hence we leave it to the future works which we mentioned in our paper. And we believe that the fundamentals of the MARCH, IDES and CAESAR algorithms could serve as the basis for the function approximation setting.
> > For the optimization part, in our algorithm, there are two sub-steps requiring optimizations. First, the opt problem in (6) is convex and we assume the optimal solution which is a very common assumption in many works (except those focusing on the optimization algorithm which is clearly not our work's focus) provided the opt problem is convex. Second, we provide a high-probability convergence guarantee for the optimization procedure to get the density ratio. What exact practical optimization stability analysis is the reviewer expecting?
> >
> > > 6. The comment 'The sample complexity in this work is almost not improved in terms of the intuitive -policy extension of the existing work' is not correct. The result of intuitive $K-$policy extension is always linearly dependent on $K$ which is very bad since it never leverages the similarity between different policies. Instead, our result is instance-dependent which incorporates the potential similarity of policies. Instance-dependent result is more refined compared with minimax results. Another significant difference between our work and the work you listed is, in there work (offline), the offline data is given with a good coverage assumption, however, in our work, we need to build such a dataset with good coverage from scratch.
> >
> > > 7. We sympathize with the reviewer’s comments regarding experimental evaluation of our algorithms. We want to reiterate that we consider the scope of our submission to be the advancement of our theoretical understanding of this problem. As such we hope to obtain a fair evaluation of our submission based on its merits to advance theoretical research in RL. That being said, we believe that the fundamentals of the MARCH, IDES and CAESAR algorithms could serve as the basis of future deep reinforcement learning algorithms equipped with efficient low order estimation procedures that can make deep exploration of novel problems more efficient than with current exploration approaches. We believe this is an exciting area of future research that lies outside the scope of the tabular results of this submission.

---

### Official Review · Reviewer_YMix · 2024-11-03

**Soundness:** 2
**Presentation:** 2
**Contribution:** 3
**Rating:** 5
**Confidence:** 4

**Summary:**

This paper explores the multiple-policy evaluation problem and introduces the CASEAR algorithm as a solution.

The algorithm begins by creating estimators for the policy visitation distributions, followed by constructing mean reward estimators for all K policies.

**Strengths:**

This paper addresses the multiple-policy evaluation problem, an area that has received relatively limited attention in the field. It is a valuable contribution to the community.

The proposed method appears to be novel.

**Weaknesses:**

The paper is difficult to follow, and my primary concern is the correctness/meaningfulness of the results.

There are numerous typos, and both the notation and overall writing could be improved.

Additional relevant references could be incorporated into this work.

**Questions:**

1. Theorem 4.8: The theorem claims that we can evaluate all K policies within an eps error margin using n trajectories. However, the paper previously only mentioned that the trajectories are drawn from some distribution mu. Without any assumptions about mu, it's easy to construct a scenario where evaluating a particular policy pi is infeasible (e.g., if mu is completely unaligned with pi). In such a case, the sample complexity upper bound would be infinite, rendering it uninformative. Clarifying when this upper bound is finite and meaningful is essential.

Can you explicitly state any assumptions made about mu and how these assumptions ensure the sample complexity bound is finite and meaningful in practice? This would help clarify an important aspect of the theoretical guarantees.

2. Comparison with Dann et al. (2023): From the discussion, it is unclear whether the bound in this paper or the one in Dann et al. (2023) is tighter.

Can you provide a more detailed comparison between the two bounds, including specific scenarios where each bound might be tighter? Can discuss the conditions under which their bound remains finite would be helpful for readers to understand the practical implications of their results?

3. While Dann’s bound is guaranteed to be finite (as d ^ max pertains to states that can be reached by some target policy), the bound in this paper can be infinite (since d pi k / mu h can be infinity).

3. Purpose of MOM: Could you clarify why the MOM method is proposed in this paper? It seems to appear abruptly. This method is typically applied to handle unbounded random variables. Is this the reason for its use here?

Can you provide a brief explanation of why MOM is necessary in their context, and how it relates to the specific challenges in their problem setting? This would help readers better understand the motivation behind using this technique.

5. The term pi det refers to all deterministic policies. The algorithm also attempts to estimate the visitation distributions for all of them (see line 381). It’s unclear why estimating only the visitation of the target policies is insufficient. Additionally, estimating the visitation distributions for all deterministic policies seems like a strong requirement.

Could you clarify why this broader estimation is necessary, and whether there are any computational or theoretical implications of this approach? Can you discuss potential alternatives or approximations that might be less computationally intensive?

6. The paper includes many terms in the form of ratios. It would be helpful to add more explanations on why these ratios are well-defined. For example. Why is hat w / hat mu bounded? Why is tilde mu / hat mu bounded?

Can you provide a brief explanation or proof for why each of these key ratios is well-defined and bounded. This would help readers better understand the technical details of the approach.

7. Is the optimization program in equation (6) well-defined? Does a solution hat mu* always exist? If the solution has an infinite value, is it still meaningful?

8. Gap-Dependent Related Works:

"Offline Reinforcement Learning Under Value and Density-Ratio Realizability: The Power of Gaps"

"Revisiting the Linear-Programming Framework for Offline RL with General Function Approximation"
Additional Related Works:

9. Some other related work:

"Offline Reinforcement Learning with Realizability and Single-Policy Concentrability"
Additional Clarifications:

"Reinforcement Learning in Low-Rank MDPs with Density Features"

10. What does w in Lemma 4.5 attempt to approximate? I guess it is d pi?

11. Why g(w h) is referred to as a gradient in the IDES algorithm?

12. Use double quotation marks (`` '') in the first paragraph.

13. Line 287 has a typo: “sum i”?

14. In Algorithm 1, D_h(s,a) = [xxx]?

15. In Lemma 4.7, the notation hat mu_1, hat mu_N  conflicts with the use of mu h in other sections.

16. mu * is unclear upon first reading in Section 3.1; a forward pointer would be helpful.

Providing a brief discussion on the existence and uniqueness of solutions to this optimization problem, as well as the implications of potentially infinite values would be great.

17. Remark 4.1 is missing "some citations".

18. Line 267 is unclear. Trajectory data is mentioned, but it states that they are drawn from mu h. Is the dataset drawn at the trajectory level or at the (state, action, next state) tuple level?

19. Can you clarify the constant C in Proposition 4.2 and explain how C h is chosen in Algorithm 1?

---

> ### Author Response · Authors · 2024-11-18
>
> Thanks for your time and feedback. We appreciate the reviewer's recognition on the novelty of our algorithm. The main concerns from the reviewer are the correctness and the meaningfulness of our work.
>
> **Meaningfulness**: As the reviewer acknowledged in the strengths part that 'this area has received relatively limited attention in the field. It is a valuable contribution to the community.', the reviewer contradicts with his/her own opinions to criticize the meaningfulness of this work.
> Nevertheless, we summarize the meaningfulness of our work as: we provide an instance-dependent sample complexity bound on multiple-policy evaluation problem which leverages the potential similarity between these policies. The algorithm adopts an innovative two-stage strategy where in the first stage, we learn an sampling distribution $\mu$ with good coverage over the target policies' space by a novel technique we call coarse estimation which requires only a low-order number of samples. In the second stage, we use that sampling distribution to evaluate all target policies by importance weighting. We not only advance our theoretical understanding of this problem. Besides, we propose many techniques and algorithms such as coarse estimation, $\beta-$distance, MARCH, IDES which can be of independent interest to be applied to other problems beyond the scope of this work.
>
> **Correctness**: For the concern of the correctness, we provided detailed answers to your questions below. Hopefully, our response can help you to better understand our work and our contributions. And we kindly request you to consider increasing your score based on the fresh comprehension of our work.
>
> We appreciate the reviewer's effort on this paper. For the questions:
> > 1. This comment is incorrect. We don't have assumptions on $\mu$ which is actually an advantage of our algorithm. However, $\mu$ is not a random distribution. $\mu$ is our learned sampling distribution (by coarse estimation and solving opt problem (6)) which has good coverage over the target policies's space. And for the boundedness of the solution to (6), we answered it in the following questions 3&6&7.
>
> > 2. We can provide instance (start from line 463) where our bound is better than their result. However, it is unclear whether ours is universally better and a comprehensive comparison is intractable since different from our result which depends on visitations (can be easily calculated given policies and MDP), their result depends on the realization of a trajectory which is hard to quantify. And to some extent, we regard our bound is preferred based on this fact.

---

> > ### Author Response · Authors · 2024-11-18
> >
> > > 3. This comment is incorrect. Our bound is guaranteed to be finite. $\mu$ is our learned sampling distribution (by coarse estimation and solving opt problem (6)) which has good coverage over the target policies's space. Besides, we also show the minimax version of our result which provides a finite upper bound under the worst case.
> >
> > > 4. The reason why we introduce MoM is to provide a high-probability result. It is not necessary. Without MoM, all our result still holds but in a manner of in-expectation.
> >
> > > 5. Before we answer this question, we want to emphasize that $\mu$ is our calculated sampling distribution which should be achievable by some behavior policy, otherwise we cannot construct such a distribution. So we need to constrain $\mu$ in a set of feasible visitations for the MDP when we solve (6).
> > The reason why we need coarse estimation of all deterministic policies is that we need a broad feasible set of $\mu$ such that by solving (6), we can claim $\mu^*$ is the optimal sampling distribution. By coarsely estimating all deterministic policies, the feasible set of $\mu$ would be the combination of all deterministic policies which covers any achievable visitation. Estimating only the visitation of the target policies is also good enough and is preferred practically. But in this case, the feasible set of $\mu$ is the combination of target policies which is relatively limited. And as we show in the paper that in some case, the optimal $\mu^*$ to (6) can lie outside the combination of target policies' visitations. Hence, for the theoretical completeness, we estimate all deterministic policies.
> > We thank this comment 'estimating the visitation distributions for all deterministic policies seems like a strong requirement.' Yes, it is challenging and our work finds a way to achieve it efficiently in terms of sample complexity. However, as we comment in Remark 4.1, the computational cost is a burden in this case. We also like this comment 'Can you discuss potential alternatives or approximations that might be less computationally intensive?'. We have the same curiosity, but unfortunately it is still an open problem which is very interesting and worth exploring. We leave it as our future direction.
> >
> > > 6. $\mu$ is not a random distribution. It is a carefully calculated optimal distribution (solution to (6)) which possesses good coverage of target policies' visitations, hence, the visitation ratios are always well-bounded. More intuitively, set $\mu'$ to be the one we specified in line 439, we can show the boundedness of the ratio, and then since $\mu'$ is a feasible solution and $\mu^*$ is the optimal solution, the ratio induced by $\mu^*$ is of course bounded as well.
> >
> > > 7. Yes, the opt problem in (6) is well defined. The boundedness can be easily verified by setting $\mu$ to be the one we specified in line 439, i.e. a bounded feasible solution.
> >
> > > 8&9. Thanks for listing these works. We'll consider to include them. In the current version, we include those most related and representative works due to the vast space of the area.
> >
> > > 10. Yes, and the answer to this question is in Lemma 4.6, $\hat w_h(s,a)/\hat \mu_h(s,a)$ is an accurate estimator of the importance weighting $d^{\pi^k}_h(s,a)/\tilde \mu_h(s,a)$.
> >
> > > 11. It is the gradient to the loss defined in line 341 which can be easily verified.
> >
> > > 12&13. Thanks for pointing out these typos. We'll update the paper to fix them.
> >
> > > 14. D_h(s,a) is the feasible set of w_h(s,a) hence it is defined by [a, b] which means we constrain w to be a$\le$w$\le$b. And it is realized in the algorithm by projection.
> >
> > > 15. Thanks for pointing it out. Yes, it is an abuse of notations. We'll fix it by updating the current version.
> >
> > > 16. Yes, we agree that a clarification on $\mu^*$ is needed to make it clear in Section 3.1 and we'll fix it. Thanks for pointing it out. The existence of the solution can be inferred by the convexity and lower-boundedness (which is zero) of the objective. For the uniqueness, we only need a solution instead of all of them, so it doesn't affect our result. For the upper-boundedness, please refer to our answer to question 7.
> >
> > > 17. Thanks for pointing it out. It is our silly mistake. The citations are as follows and we'll include them in the updated version.
> > [1]. Harnessing Density Ratios for Online Reinforcement Learning.
> > [2]. Towards Optimal Regret in Adversarial Linear MDPs with Bandit Feedback.
> >
> > > 18. It is sampled at the trajectory level. For each $\mu_h$, we sample a bunch of trajectories such that the distribution of these trajectories at layer h is $\mu_h$. And that's why our result has a dependency on $H^4$ since we need to sample trajectories for each $\mu_h$ respectively.
> >
> > > 19. $C$ in Proposition 4.2 is a universal constant which bounds the performance of the estimator. We don't need to choose $C$ in our algorithm.

---

> > > ### Comment · Reviewer_YMix · 2024-11-29
> > >
> > > I'll keep my scores for now. Thank the authors for the comments. I have read them and also checked reviews from others.

---

### Official Review · Reviewer_GyRZ · 2024-11-03

**Soundness:** 2
**Presentation:** 2
**Contribution:** 2
**Rating:** 5
**Confidence:** 5

**Summary:**

The paper addresses the problem of multiple-policy evaluation in reinforcement learning, where the goal is to estimate the performance of a set of $K$ target policies up to a specified accuracy with high probability. The authors propose a novel, sample-efficient algorithm called **CAESAR** (Coarse and Adaptive EStimation with Approximate Reweighing) to tackle this problem. CAESAR operates in two phases: first, it coarsely estimates the visitation distributions of the target policies using the **MARCH** (Multi-policy Approximation via Ratio-based Coarse Handling) algorithm; second, it approximates the optimal offline sampling distribution and computes importance weighting ratios using the **IDES** (Importance Density Estimation) algorithm. The authors provide theoretical guarantees for their algorithm, demonstrating a non-asymptotic, instance-dependent sample complexity of $\tilde{O}\left( \frac{H^4}{\epsilon^2} \sum_{h=1}^H \max_{k \in [K]} \sum_{s,a} \frac{(d_h^{\pi_k}(s,a))^2}{\mu^*_h(s,a)} \right)$, where $H$ is the horizon, $\epsilon$ is the desired accuracy, $d_h^{\pi_k}$ is the visitation distribution of policy $\pi_k$, and $\mu^*_h$ is the optimal sampling distribution. The paper also discusses how CAESAR compares to existing methods and its applicability to policy identification.

**Strengths:**

- **Novel Approach**: The paper introduces the CAESAR algorithm, which combines coarse estimation and importance weighting to achieve sample-efficient multiple-policy evaluation. This novel approach addresses a gap in existing reinforcement learning literature.

- **Theoretical Guarantees**: The authors provide rigorous theoretical analysis, offering non-asymptotic, instance-dependent sample complexity bounds for their algorithm. This strengthens the validity of their contributions.

- **Innovative Sub-algorithms**: The MARCH algorithm for coarse estimation of visitation distributions and the IDES algorithm for importance density estimation are innovative and could have applications beyond this work. They effectively adapt and extend existing techniques to the finite-horizon setting.

- **Discussion and Comparisons**: The paper includes a thorough discussion section that compares CAESAR with existing methods, highlighting its advantages and situating it within the broader research landscape.

**Weaknesses:**

- **Clarity and Readability**: The presentation lacks clarity in several sections. The theoretical analysis is particularly dense, with complex notation and insufficient intuitive explanations, which may impede understanding for readers not deeply familiar with the topic.

- **Practical Implementation**: The paper assumes access to an oracle for solving convex optimization problems, which may not be practical in real-world applications. The computational complexity and practical feasibility of implementing CAESAR are not thoroughly discussed.

- **Lack of Empirical Evaluation**: There is a significant lack of empirical experiments or simulations to validate the theoretical findings. Including empirical results would strengthen the paper by demonstrating the practical effectiveness of the proposed algorithms.

- **Scalability Concerns**: While the MARCH algorithm aims to estimate visitation distributions efficiently, the scalability of the method to environments with large state and action spaces is not fully addressed.

- **Comparison with Prior Work**: Although the paper discusses existing methods, it lacks comprehensive empirical comparisons. Demonstrating how CAESAR performs relative to prior algorithms in practice would enhance the contribution.

**Questions:**

1. **Intuitive Explanation of CAESAR**: Could the authors provide more intuitive explanations or illustrative examples to clarify the key steps of the CAESAR algorithm, particularly how coarse estimation contributes to sample efficiency?

2. **Computational Complexity**: How does the computational complexity of CAESAR compare with existing methods, especially considering the need to solve convex optimization problems and estimate importance weights? Are there practical algorithms that can implement CAESAR efficiently without reliance on an oracle?

3. **Empirical Validation**: Have the authors considered conducting empirical evaluations of CAESAR on benchmark reinforcement learning tasks to demonstrate its practical performance and verify the theoretical claims?

4. **Scalability to Large Spaces**: How does the proposed method scale to environments with large or continuous state and action spaces? Are there strategies to mitigate potential computational bottlenecks in such scenarios?

5. **Assumptions and Limitations**: Can the authors discuss the practicality of the assumptions made in the theoretical analysis and how they might impact real-world applications? Are there ways to relax these assumptions while maintaining the algorithm's performance guarantees?

---

> ### Author Response · Authors · 2024-11-18
>
> Thanks for your time and feedback. We appreciate your acknowledgement on the novelty of our work and the adaptivity of our techniques to other applications. We provide careful responses to the reviewer's concerns below and we kindly request the reviewer to consider increasing the score based on our clarifications.
>
> For the weaknesses & questions:
> > **Intuitive Explanation**: The key step of our algorithm is the coarse estimation with the intuitive explanation as follows. We want to build a sampling distribution $\mu$ which has a good coverage of all target policies' space (by solving (4)). However, since we don't know the true visitations ($d^{\pi^k}$) of these target policies, we cannot learn such a sampling distribution. Hence, we propose to coarsely estimate the visitations of the target policies first. With these coarse estimators ($\hat d^{\pi^k}$), we convert the problem (4) with unknown quantities to a problem (6) with known quantities hence solvable. And we show that although these estimators are coarse (up to multiplicative constant), it is enough in the sense that the optimal sampling distribution $\tilde \mu^*$ calculated with the help of coarse estimators are close to the true optimal sampling distribution $\mu^*$ which is the solution to (4). And importantly, this procedure of coarse estimation only costs $\tilde{O}(1/\epsilon)$ samples which is low-order and negligible for the whole problem.
>
> > The full procedure of our method can be summarized as: first, we explore in the MDP and figure out an optimal sampling distribution (i.e. an optimal behavior policy) by coarse estimation using only few samples; second, we just collect trajectories by implementing this behavior policy (i.e. to construct our learned optimal sampling distribution) and use these samples to evaluate the performance of the target policies.

---

> > ### Author Response · Authors · 2024-11-18
> >
> > > **Readability**: We are sorry that the presentation and organization pose difficulty to some of our readers of understanding our stuff. We will take an iteration to make it more accessible and fix existing typos. This can be done quickly in preparing a camera-ready version. And it would be a pity to reject a work because it is theory-intensive.
> >
> > > **Computational Complexity**: Assuming a solution to a well-defined convex opt problem is acceptable. The current optimization solvers can give such a solution within polynomial time with respect to the number of constraints of the opt problem. Besides, our result can easily be extended to tolerate the opt error. In fact, the computational burden comes from the large feasible space of $\mu$ (i.e. the constraints). The reason why we adopt such a large feasible space of $\mu$ which actually contains all feasible visitations of the MDP is for the theoretical completeness. We want to show $\mu^*$ is optimal over all possible distributions. Practically, we can just constrain $\mu$ to be within the combination of target policies which is good enough. And we leave the problem of finding an efficient low-order exploration algorithm for deep RL as an interesting future direction which is way beyond the scope of this paper.
> >
> > > **Scalability**: Our algorithm is scalable in terms of sample complexity since our sample complexity is upper bounded by a quantity which is just polynomially dependent on state-action space (see line 447). And this is the best can be achieved in tabular case. For the concerns of scalability in terms of computational complexity, please refer to our responses above. Extending our work to function approximation setting would be an important step to achieve scalability over real-world scenarios. We believe the techniques developed in our paper including the coarse estimation, $\beta-$distance and the pipeline of finding an approximate optimal sampling distribution at a low-order cost could inspire the solution to this problem under the function approximation setting.
> >
> > > **Empirical Evaluation**: We sympathize with the reviewer’s comments regarding experimental evaluation of our algorithms. **We want to reiterate that we consider the scope of our submission to be the advancement of our theoretical understanding of this problem. As such we hope to obtain a fair evaluation of our submission based on its merits to advance theoretical research in RL**. That being said, we believe that the fundamentals of the MARCH, IDES and CAESAR algorithms could serve as the basis of future deep reinforcement learning algorithms equipped with efficient low order estimation procedures that can make deep exploration of novel problems more efficient than with current exploration approaches. We believe this is an exciting area of future research that lies outside the scope of the tabular results of this submission.
> >
> > > **Assumptions**: Basically, our work doesn't require any strict assumption. We just assume a finite state and action space since we are talking about the tabular setting. And we assume a non-negative bounded reward which is an almost universal setting in RL works.
> >
> > > **Limitations and Contributions**: We listed the limitations clearly in our paper and also pointed out possible future directions. These include the sub-optimal dependency on $H$, a possible more instance-dependent result such as variance-dependent and reward-dependent bounds, the lack of efficient low-order exploration method for deep RL and the extension to function approximation setting. **However, we don't think these limitations invalidate or obscure the contributions of this work**. We proposed many techniques that can be of independent interest which is also acknowledged by the reviewer. We solve the problem under the tabular case which is non-trivial and actually requires lots of smart designs, and we pave the theoretical foundation upon which further works will build to step towards more general and practical settings.

---

### Official Review · Reviewer_cv8X · 2024-11-04

**Soundness:** 2
**Presentation:** 2
**Contribution:** 2
**Rating:** 3
**Confidence:** 4

**Summary:**

This paper introduces CAESAR, a sample-efficient algorithm for multiple-policy evaluation in reinforcement learning. The approach involves computing an approximately optimal sampling distribution and using data drawn from this distribution to evaluate multiple policies simultaneously. The authors also provide a theoretical analysis of the algorithm’s sample complexity.

**Strengths:**

1. The work is specifically designed to handle multiple-policy evaluation efficiently, avoiding the linear scaling in sample complexity that would occur if each policy were evaluated independently.

**Weaknesses:**

1. Although the authors say that their work is practical, they have not shown any experimental results of their algorithm. From a theoretical perspective, they also didn't show whether their sample complexity bound is tight enough, compared with existing works.

2. This paper has some funny typos and is not well-organized. (1) In line 313, they write "(some citations)" without really citing anything. (2) In line 279, the $\mu$ is not defined in their algorithm's input. (3) $\alpha^*_\pi$ in line 383 is not clearly defined, etc. Besides these typos, so many subroutines and abbreviations also make this paper very hard to understand. The authors put too many trivial lemmas in their main text, making their main contribution unclear.

3. In this work, the authors propose two algorithms, MARCH and IDES, as subroutines of their main algorithm, CAESAR. MARCH provides a coarse estimation of the visitation distribution for all deterministic policies. And IDES rely on the estimated visitation distribution from MARCH. Thus, their main algorithm CAESAR works only on deterministic policies. Confusely, in the preliminary section, the authors define policies as stochastic policies. However, throughout the paper, they didn't make justifications on why their method shifts to be only applicable to deterministic policies.

4. As discussed above, the MARCH algorithm provides coarsely estimated visitation distributions of the target policies, which introduces errors. Likewise, the IDES algorithm relies on a sampling distribution derived from an approximate optimization problem (6), which minimizes only an upper bound of the estimation variance rather than the exact variance. These approximations result in a biased estimator, which is not preferable.

5. In Session 5, when comparing their sample complexity with related works, the authors apply the CR-lower bound (line 419) to derive a lower bound of their method CAESAR. However, as the authors themselves point out, the CR-lower bound is applicable only to unbiased estimators. However, as far as I understand, the estimator from CAESAR is not guaranteed to be unbiased. This discrepancy makes the lower-bound discussion less reliable.

6. The authors introduce the MARCH algorithm to estimate the visitation distributions of target policies; however, they didn't compare it to any existing algorithms designed for the same purpose.

**Questions:**

1. The authors claim that their method is fully offline. However, I am not sure if their IDES algorithm can work without online interaction. Namely, how is the $s^{i'} _h$ generated without interacting with the environment?

2. In Lemma 4.5, it is clear that the errors propagate step-by-step, with the error of each time step twice the error in the previous step. How can this eventually lead to an accurate estimation?

---

> ### Author Response · Authors · 2024-11-19
>
> Thanks for your time and feedback. **Unfortunately we feel that there are some misunderstandings on our work by the reviewer which may hinder the reviewer to correctly perceive the contributions of this work**. We carefully respond to the reviewer's concerns below. Hopefully, these responses may help the reviewer understand our work better. And we kindly request the reviewer to consider increasing the score based on the fresh comprehension of our work.
>
> Here we give a brief list of clarifications on the points that the reviewer shows misunderstandings of our work and we will explain in details below.
> 1. MARCH is **not** used to coarsely estimate the visitation distributions of target policies.
> 2. IDES **doesn't** rely on the estimated visitation distribution from MARCH.
> 3. Our algorithm is **not** fully offline.
>
> Among several comments (3&4&6) in the weaknesses part, the reviewer keeps the opinion that we introduce MARCH algorithm to estimate the visitation distributions of target policies. We want to emphasize here that this comment is incorrect. MARCH is implemented to coarsely estimate the visitation distributions of all deterministic policies while the target policies are coarsely estimated independently (refer to Proposition 4.2). Hence, the target policies can be stochastic.
>
> MARCH produces coarse visitation estimators for all deterministic policies. And when solving (6), we can constrain the sampling distribution $\mu$ to be in the combination of all deterministic policies, i.e. the feasible set of $\mu$ is the one containing all achievable distributions in the MDP, otherwise, the optimal sampling distribution is either unachievable by any behavior policy or sub-optimal. IDES uses samples from our learned optimal distribution $\mu$, instead of depending on the coarse estimator for all deterministic policies.
>
> Our algorithm is not offline. Our method consists of two stages. In the first stage, we collect few samples (which is low-order and negligible) to coarsely estimate the visitation distributions of target policies as well as all deterministic policies. With these coarse estimators, we can solve (6) which gives us an achievable good sampling distribution for the meta problem. Then in the second stage, we use trajectories from this sampling distribution to perform the evaluation of all target policies simultaneously.

---

> ### Author Response · Authors · 2024-11-19
>
> Responses to the weaknesses:
> > 1. Empirical Experiments: We sympathize with the reviewer’s comments regarding experimental evaluation of our algorithms. We want to reiterate that we consider the scope of our submission to be the advancement of our theoretical understanding of this problem. As such we hope to obtain a fair evaluation of our submission based on its merits to advance theoretical research in RL. That being said, we believe that the fundamentals of the MARCH, IDES and CAESAR algorithms could serve as the basis of future deep reinforcement learning algorithms equipped with efficient low order estimation procedures that can make deep exploration of novel problems more efficient than with current exploration approaches. We believe this is an exciting area of future research that lies outside the scope of the tabular results of this submission.
>
> > Tightness of the Bound: Unfortunately we cannot show if it is the tightest bound one can achieve for the multi-policy evaluation problem. As we write in the paper as one of our limitations (line 431) the lower bound of this problem is super challenging which is beyond the scope of this work and we leave it to future works.
> Nevertheless, it doesn't invalidate or obscure the main contributions of our work which we summarize as follows: we provide an instance-dependent sample complexity bound on multiple-policy evaluation problem which leverages the potential similarity between these policies. This bound enjoys lots of benefits compared with the existing one such as the interpretability (easy to quantify the difficulty of the problem when given the MDP and policy), the improvement of sample complexity in some special cases. We propose an innovative technique named coarse estimation to approximately learn the optimal sampling distribution at a cost of only a low-order number of samples. We propose IDES to accurately estimate importance ratios which extends DualDICE to episode MDPs and the extension is not trivial. Conclusively, we not only advance the theoretical understanding of this problem. Besides, we propose many techniques and algorithms such as coarse estimation, distance, MARCH, IDES which can be of independent interest to be applied to other problems beyond the scope of this work.
>
> > 2. Thanks for pointing them out. We feel sorry that there exist some typos in the presentation. We admit these are our negligence. And we will fix them through a quick iteration.
>
> > 3.  As we have clarified above, the target policies are not coarsely estimated by MARCH. And IDES doesn't rely on the estimated visitation distribution from MARCH. The target polices can be stochastic.
>
> > 4. That's why the coarse estimation is amazing. Coarse estimation is an innovative and powerful technique we proposed for this problem and can be of independent interest. By only paying a low-order number of samples ($\tilde O(1/\epsilon)$), we can estimate the distribution up to multiplicative constant.  Although, the estimator is coarse and the opt problem (6) is just an approximation to the true problem (4), we theoretically shows that the error between the approximately optimal sampling distribution $\tilde \mu^*$ and the true optimal distribution $\mu^*$ is also up to multiplicative constant which is acceptable.
> For the bias of our estimator, we don't think it is a problem in terms of the quality of the estimator. We measure the accuracy of our estimator by the L1 distance, i.e. $|\hat V -V|\le \epsilon$. And our result holds with high probability. We can set $\epsilon$ to any value that meets our need.
>
> > 5. We agree with the reviewer that our final performance estimator is biased which makes the direct application of CR-lower bound on our case incorrect. Thanks a lot for pointing this out. We will update our paper. However, as we illustrated above in the tightness part, this limitation does not invalidate our main contributions.
>
> For the questions:
> > 1. We have clarified above that our algorithm is not offline. We don't agree that we claim our method is fully offline in anywhere of the paper. We are confused at where the reviewer finds this claim.
> Besides, the offline setting means one obtains samples from a given fixed distribution. Our second stage (including IDES) is conducted in an offline manner where we sample data from the fixed sampling distribution $\tilde\mu^*$ which is learned in the first stage.
>
> > 2. This comment 'with the error of each time step twice the error in the previous step.' is inaccurate. The error of each timestep is the summation of the opt error of the current step and the error accumulated before. The additive error propagation is acceptable and easy to control by setting opt error at each timestep as $\epsilon/H^2$. The polynomial dependency on $H$ is acceptable in our scenario.

---

### Meta-Review · Area_Chair_ksFF · 2024-12-21

**Metareview:**

This paper studies multiple-policy evaluation in reinforcement learning, introducing the CAESAR algorithm, which employs coarse estimation and importance weighting to achieve sample-efficient policy evaluation. While the work provides theoretical sample complexity bounds and a novel two-stage approach, it lacks empirical validation and an information-theoretical lower bound to establish the optimality of the method. Reviewers noted issues with presentation clarity, connections to offline policy evaluation, and incomplete discussions on practical feasibility and scalability. The AC acknowledges the authors' concerns about reviewer misunderstandings, but these do not significantly impact the main criticisms regarding the lack of a lower bound or practical validation. These unresolved issues led to the recommendation for rejection.

**Additional Comments On Reviewer Discussion:**

During the rebuttal period, reviewers raised several concerns: (1) the lack of empirical evaluation, (2) unclear presentation of the algorithm’s theoretical contributions, (3) missing comparisons with key related works, (4) practical feasibility issues, including computational complexity, and (5) insufficient motivation and applicability for the multi-policy evaluation framework. The authors provided detailed responses, clarifying their theoretical claims, addressing misunderstandings regarding offline versus online settings, and offering a broader discussion of their techniques' potential applicability. However, they acknowledged some limitations, such as the absence of empirical results and unresolved scalability concerns, while promising improvements in future iterations. While the AC acknowledges the authors' concerns regarding reviewer misunderstandings, these do not affect the core critiques of the paper, including the lack of lower bounds and empirical validation.

---

### Decision · Program_Chairs · 2025-01-22

Reject